# Transformers as Intrinsic Optimizers: Forward Inference through the Energy Principle

## Abstract

Transformers have demonstrated strong adaptability across a wide range of tasks and become the backbone of modern Large Language Models (LLMs). However, their underlying mechanisms remain open for further exploration. The energy-based perspective has long provided a valuable principle for understanding neural computation. In this paper, we revisit the energy principle as a framework for understanding attention-based Transformers. Within the proposed framework, standard attention can be viewed as a special case of minimizing the Helmholtz free energy when the energy function takes the form of elastic potential energy, with residual connections ensuring that this optimization proceeds in an incremental manner. Building on this connection, we incorporate the forward pass and parameter updates during model training into a unified alternating optimization perspective where parameter updates follow conventional training objectives while the model architecture is responsible for locally optimizing on the energy-based regularization. Furthermore, we extend the first-order energy update of standard attention to a second-order form based on Newton's method, which ultimately introduces a covariance matrix to precondition the update directions of tokens. Meanwhile, we extend the above analysis to the multi-head case, where energy minimization is performed across multiple low-dimensional subspaces. Our experiments provide preliminary support for the potential of using the energy-based framework to design attention mechanisms.

## 1 Introduction

Energy-based formulations have long underpinned theories of neural computation and the modeling of neural networks (Hopfield, 1982; Ackley et al., 1985; LeCun et al., 2006). One of the most influential works applying the concept of energy to pattern recognition is Associative Memory models, also known as Hopfield Networks Hopfield (1982; 1984), which implement associative memory by defining an energy function over neuron states. Modern Hopfield Networks have been largely enhanced to achieve greater storage capacity through the design of new energy functions (Krotov & Hopfield, 2016; Ramsauer et al., 2020; Krotov, 2023). Additionally, based on the energy concept, LeCun et al. (2006) propose Energy-Based Models (EBMs) as a unifying framework for learning, where the training objective is to assign low energy to plausible configurations of variables and high energy to implausible ones. In fact, many modern self-supervised learning (SSL) methods can be naturally interpreted within this framework (Chen et al., 2020; He et al., 2020; LeCun, 2022; Gladstone et al., 2025). The energy-based perspective has demonstrated great appeal in the development of deep neural networks.

On the other hand, in recent years, with the development of the SSL paradigm, pretrained large language models (LLMs) have achieved remarkable success across various areas (Kenton & Toutanova, 2019; Brown et al., 2020). This success is not only attributed to these effective paradigms such as autoregressive training but also relies on the Transformer-based architecture as the foundational backbone (Vaswani et al., 2017). Therefore, many studies have begun to explore the theoretical mechanisms underlying the Transformer architecture, with a popular approach being to connect the model architecture to unrolled optimization (Gregor & LeCun, 2010; Tolooshams & Ba, 2021; Chan et al., 2022; Hinton, 2022). Zhou et al. (2022) explained that stacked self-attention modules can promote grouping and noise filtering using the information bottleneck principle. Yu et al. (2024b) showed that Transformer-like deep network layers can naturally be connected to an optimization

process aimed at sparse rate reduction. Wang et al. (2025b) pointed out that compressing noisy to-ken representations and the corresponding denoising operations can naturally give rise to the form of multi-head self-attention. Actor et al. (2025) showed that optimizing latent features in multinomial regression align with dynamics induced by the attention blocks.

In addition to above explanations, some works have also attempted to establish a connection between energy-based principles and Transformers. For example, Ramsauer et al. (2020) proposed a modern Hopfield network whose energy objective corresponds to an update rule that takes a form similar to the attention mechanism in Transformers. Furthermore, Hoover et al. (2023) proposed the Energy Transformer which integrates multi-head energy attention with a Hopfield Network module and demonstrated good empirical performance across various tasks. Although these studies establish certain connections between energy and Transformers, the design of energy functions is often not straightforward and lacks a unified framework to understand, which limits both our understanding of Transformers and the potential design of model architectures.

In this paper, we revisit the principle of energy to view attention-based Transformer models. Our work mainly follows the following line of presentation:

**(i.) Energy Framework for Attention.** We first present an energy framework to provide a princi-pled understanding of attention-based models. Within this framework, standard attention emerges as a special case where the global energy $F^*$ and the energy function $E_i$ take the forms of Helmholtz free energy and elastic potential energy respectively. From this perspective, the forward inference of standard attention corresponds to performing first-order gradient descent (GD) to minimize the free energy, with residual connections ensuring that the update is carried out in an incremental manner.

**(ii.) Unified Alternating Optimization Perspective.** Building on this connection, we point out that both the forward computation and the parameter updates in Transformer training can be incorpo-rated into a unified alternating optimization perspective: parameter updates follow the conventional training objectives, while the forward pass is responsible for local optimization of the regularization term which is determined by the model architecture itself and carried out in the form of free energy.

**(iii.) Second-order Attention Updates.** Furthermore, we propose that the attention structure can be modified based on this energy-based framework. Specifically, we extend the local energy descent that is originally based on first-order GD to a second-order form grounded in Newton's method and then employ a Taylor expansion approximation to reduce its computational cost to the same order as standard attention. Compared to standard attention, the induced new attention for uses the covariance matrix to precondition the original update directions, allowing tokens to adaptively adjust their movements along different dimensions.

**(iv.) Extension to Multi-head Case.** Meanwhile, we extend the above analysis to the multi-head attention case whose forward computation can be viewed as optimizing the average Helmholtz free energy across multiple low-dimensional manifolds. We also apply the second-order GD update to modify the multi-head attention and the resulting induced model is named MHA2nd1st, which also uses the covariance matrix to adjust the updates within each subspace. Our experiments offer preliminary support for the effectiveness of the newly induced attention structure.

## 2 HELMHOLTZ FREE ENERGY AS A PRINCIPLE FOR ATTENTION

### 2.1 CONNECTING ATTENTION WITH HELMHOLTZ FREE ENERGY

The attention mechanism in Transformers is designed to model the interactions between tokens. For a given input $z \in \mathbb{R}^d$, we assume that the set of tokens[1] interacting with it is $\{h_i\}_{i=1}^N \in \mathbb{R}^{d \times N}$. The output of the standard attention layer in the single-head case can be formalized as[2]

$$\text{Atten}(z) = z + W_V H \text{softmax}\left(H^T W_K^T W_Q z\right) = z + \sum_{i=1}^N \frac{e^{z^T W_Q^T W_K h_i / T}}{Z'} W_V h_i, \quad (1)$$

---

[1]Here we do not impose any restrictions on the attention setup. For example, in the causal setting (decoder), $z$ can be the token at position $N + 1$, that is, $z = h_{N+1}$, while $\{h_i\}_{i=1}^N$ denotes the $N$ preceding tokens; in the bidirectional setting (encoder), $z$ can be the token at any given position while $\{h_i\}_{i=1}^N$ are remaining ones.

[2]Here, for simplicity of notation, we absorb the factor $1/\sqrt{d}$ into the parameters.

where $\boldsymbol{H} = [\boldsymbol{h}_1, \boldsymbol{h}_2, ..., \boldsymbol{h}_N] \in \mathbb{R}^{d \times N}$, $T$ is the temperature and $\boldsymbol{W}_V, \boldsymbol{W}_K, \boldsymbol{W}_Q \in \mathbb{R}^{d \times d}$ are learnable parameters. In addition, $Z' = \sum_{j=1}^{N} e^{\boldsymbol{z}^T \boldsymbol{W}_Q^T \boldsymbol{W}_K \boldsymbol{h}_j / T}$ is the normalizing term.

To illustrate how the Transformer connects to the optimization objective of minimizing the Helmholtz free energy, we can first regard each token as a particle, with multiple particles together forming a system. We assume that there are already $N$ particles within our system, and the position of the $i$-th particle in the system can be denoted by $\boldsymbol{h}_i \in \mathbb{R}^d$. We want to place a new particle into the system with its position denoted by $\boldsymbol{z} \in \mathbb{R}^d$ and the other particles will exert interactions on it thereby generating the potential energy. The energy exerted on the new particle by the $i$-th particle can be denoted as $E(\boldsymbol{z}, \boldsymbol{h}_i)$ and we also use $E_i$ for simplification.

We define the internal energy of the system (respect to $\boldsymbol{z}$) as $U = \sum_{i=1}^{N} p_i E_i$ where $p_i > 0$ is the assigned weight to the $i$-th particle and satisfies $\sum_{i=1}^{N} p_i = 1$. Furthermore, the entropy of the system can be represented as $S = -\sum_{i=1}^{N} p_i \log p_i$. The free energy of the system is the portion of its internal energy that is not consumed by disorder, that is,

$$F = U - TS = \sum_{i=1}^{N} p_i E_i + T \cdot \sum_{i=1}^{N} p_i \log p_i, \tag{2}$$

where $T$ is the temperature characterizing how much the internal energy is unavailable due to disorder (entropy). We first show that when the weights $p_i$ follow the Boltzmann distribution, the system's free energy will reach its minimum:

**Lemma 1** (Helmholtz free energy). *Define the partition function as $Z = \sum_{i=1}^{N} e^{-E_i/T}$. The system's free energy defined by Eq (2) attains its minimum value*

$$F^* = -T \log Z = -T \log \sum_{i=1}^{N} e^{-E_i/T}, \tag{3}$$

*when $p_i$ satisfies the Boltzmann distribution, i.e., $p_i = \frac{e^{-E_i/T}}{Z}$.*

The proof can be seen in Appendix A.2. We next show that the forward inference of attention defined in Eq.(1) can be interpreted optimizing the Helmholtz free energy in a special case where the energy function takes the form of an elastic potential parameterized by $\boldsymbol{W}$ and the particles mapped by $\boldsymbol{W}$ are constrained to lie on a hypersphere.

**Theorem 1.** *Let the energy function $E_i = E(\boldsymbol{z}, \boldsymbol{h}_i)$ take the parameterized elastic potential form, that is,*

$$E_{\boldsymbol{W}}(\boldsymbol{z}, \boldsymbol{h}_i) = \frac{1}{2} \|\boldsymbol{z} - \boldsymbol{W} \boldsymbol{h}_i\|^2,$$

*where $\boldsymbol{W} \in \mathbb{R}^{d \times d}$ is the learnable parameter. Then the Helmholtz free energy can be formalized as*

$$F^* = -T \log \sum_{i=1}^{N} e^{-\frac{\|\boldsymbol{z} - \boldsymbol{W} \boldsymbol{h}_i\|^2}{2T}}. \tag{4}$$

*Assume that $\boldsymbol{z}$ and all $\boldsymbol{W} \boldsymbol{h}_i$ lie on a hypersphere of radius $\rho$, that is, $\|\boldsymbol{z}\| = \|\boldsymbol{W} \boldsymbol{h}_i\| = \rho$ for all $i \in [N]$. Then the forward inference of the standard attention defined in Eq (1) can be modeled as one gradient descent step for minimizing $F^*$ with the learning rate $\eta$ when setting $\boldsymbol{W}_Q^T \boldsymbol{W}_K = \boldsymbol{W}$ and $\boldsymbol{W}_V = \eta T \boldsymbol{W}$.*

*Proof.* Using the assumption that $\|\boldsymbol{z}\| = \|\boldsymbol{W} \boldsymbol{h}_i\| = \rho$ for all $i \in [N]$, we first have

$$F^* = -T \log \sum_{i=1}^{N} e^{-\frac{\|\boldsymbol{z} - \boldsymbol{W} \boldsymbol{h}_i\|^2}{2T}} = -T \log \sum_{i=1}^{N} e^{\frac{\boldsymbol{z}^T \boldsymbol{W} \boldsymbol{h}_i}{T}} + \rho^2 = \tilde{F}^* + \rho^2$$

where $\tilde{F}^* = -T \log \sum_{i=1}^{N} e^{\frac{\boldsymbol{z}^T \boldsymbol{W} \boldsymbol{h}_i}{T}}$. We can take the derivative of $F^*$ with respect to $\boldsymbol{z}$ to obtain

$$\nabla_{\boldsymbol{z}} F^* = \nabla_{\boldsymbol{z}} \tilde{F}^* = -T \nabla_{\boldsymbol{z}} \log \sum_{i=1}^{N} e^{\frac{\boldsymbol{z}^T \boldsymbol{W} \boldsymbol{h}_i}{T}} = -T \sum_{i=1}^{N} \frac{e^{\boldsymbol{z}^T \boldsymbol{W} \boldsymbol{h}_i / T}}{Z} \boldsymbol{W} \boldsymbol{h}_i,$$

Table 1: Comparison of different attention forms under the energy-based framework.

| Global Energy $F^*$ | Energy function $E_i$ | GD Form | Induced Attention |
|---|---|---|---|
| $-\frac{T}{2}\sum_i E_i^2$ | $-\boldsymbol{z}^T\boldsymbol{W}\boldsymbol{h}_i$ | First-order GD | Linear Attention |
| $-T\log\sum_i e^{-E_i/T}$ | $\frac{1}{2}\|\boldsymbol{z}-\boldsymbol{W}\boldsymbol{h}_i\|^2$ or $-\boldsymbol{z}^T\boldsymbol{W}\boldsymbol{h}_i$ | First-order GD | Standard Attention |
| $-T\log\sum_i e^{-E_i/T}$ | $\frac{1}{2}\|\boldsymbol{z}-\boldsymbol{W}\boldsymbol{h}_i\|^2$ or $-\boldsymbol{z}^T\boldsymbol{W}\boldsymbol{h}_i$ | Newton's Method | Atten2nd (Section 3) |

where $Z = \sum_{j=1}^N e^{\boldsymbol{z}^T\boldsymbol{W}\boldsymbol{h}_j/T}$. Then, given an initial value $\boldsymbol{z}^{(0)}$, we can apply gradient descent to minimize the objective $F^*$. Suppose the learning rate is $\eta$, the iteration is given by

$$\boldsymbol{z}^{(k+1)} = \boldsymbol{z}^{(k)} - \eta\nabla_{\boldsymbol{z}^{(k)}}F^* = \boldsymbol{z}^{(k)} + \sum_{i=1}^N \frac{e^{(\boldsymbol{z}^{(k)})^T\boldsymbol{W}\boldsymbol{h}_i/T}}{Z}\eta T\boldsymbol{W}\boldsymbol{h}_i.$$

By comparing with Eq (1), we can rewrite the learnable $\boldsymbol{W}$ as $\boldsymbol{W} = \boldsymbol{W}_Q^T\boldsymbol{W}_K$ and further set $\eta T\boldsymbol{W} = \boldsymbol{W}_V$. Then, we will have $Z' = Z$ and the above equation can be reformulated as

$$\boldsymbol{z}^{(k+1)} = \text{Atten}(\boldsymbol{z}^{(k)}) = \boldsymbol{z}^{(k)} + \sum_{i=1}^N \frac{e^{(\boldsymbol{z}^{(k)})^T\boldsymbol{W}_Q^T\boldsymbol{W}_K\boldsymbol{h}_i/T}}{Z}\boldsymbol{W}_V\boldsymbol{h}_i,$$

which has the same form as the attention layer in Eq (1). Thus, we complete our proof. □

Below, we discuss Theorem 1 from the following perspectives.

**(i.) Specific selection and constraint on the energy function.** First, we note that in Theorem 1, the energy function takes a form similar to elastic potential energy $E_i = \frac{1}{2}k\Delta^2$ where $\Delta = \|\boldsymbol{z} - \boldsymbol{h}_i\|$ and the elastic constant $k = 1$, meaning that when a particle (token) $\boldsymbol{z}$ deviates from the existing $\boldsymbol{h}_i$, it will be pulled back toward the position of $\boldsymbol{h}_i$ [3]. Ultimately, when $\boldsymbol{z} = \boldsymbol{h}_i$, the new particle $\boldsymbol{z}$ will be in a stable state with minimal energy $E(\boldsymbol{z}, \boldsymbol{h}_i) = 0$. These pulling forces ensure that $\boldsymbol{z}$ maintains the semantic similarity with all existing tokens. Furthermore, to make the energy function more flexible, we parameterize it as a learnable function, that is, $E_i = E_{\boldsymbol{W}}(\boldsymbol{z}, \boldsymbol{h}_i) = \frac{\|\boldsymbol{z}-\boldsymbol{W}\boldsymbol{h}_i\|^2}{2}$ where $\boldsymbol{W} \in \mathbb{R}^{d\times d}$ is the learnable parameters.

In addition, we also impose the constraint on the norms of $\boldsymbol{z}$ and $\boldsymbol{W}\boldsymbol{h}_i$, requiring them to lie on a hypersphere of fixed radius $\rho$. In practice, we often use techniques like LayerNorm (Ba et al., 2016) or RMSNorm (Zhang & Sennrich, 2019) to allow more flexible adjustment of these norms. When this constraint is relaxed so that $\boldsymbol{z}$ and the projected $\boldsymbol{h}_i$ lie within the sphere of radius $\rho$, we will have $F^* \leq \tilde{F}^* + \rho^2$ and the forward inference of attention will optimize the upper bound $\tilde{F}^*$ instead of $F^*$ directly. In fact, $\tilde{F}^*$ can also be viewed as the Helmholtz free energy in the case where $E_i = -\boldsymbol{z}^T\boldsymbol{W}\boldsymbol{h}_i$.

**(ii.) Extension to a more general Energy-based framework.** In fact, the above special case can be extended to a more general energy-based framework, which is described in Table 1. This framework consists of three key components: the global energy $F^*$, the energy function $E_i$, and the gradient descent (GD) algorithm applied. When different modifications are made to these components, corresponding attention architectures will be naturally induced. For example, when $F^*$ is taken in a quadratic-sum form, we obtain the linear attention formulation (see Appendix A.3). This framework not only provides insights into understanding existing attention mechanisms but also facilitates the design of new variants. For example, when higher-order optimization methods (e.g., Newton's method) are employed, novel attention forms will naturally emerge (see Section 3).

---

[3]We also note that in this special chosen of $E_i$, each term (also called Boltzmann factor) in the partition function takes the form of a radial basis function (RBF), that is, $\exp(-E_i/T) = \exp(-\|\boldsymbol{z} - \boldsymbol{h}_i\|^2/2T)$. These terms are also approximated by the kernel mapping functions (Choromanski et al., 2020; Katharopoulos et al., 2020), that is, $\exp(-\|\boldsymbol{z} - \boldsymbol{h}_i\|^2/2) = \phi(\boldsymbol{z})^T\phi(\boldsymbol{h}_i)$. Thus the free energy can also be written as $F^* = -T\log\sum_i \phi(\boldsymbol{z})^T\phi(\boldsymbol{h}_i)$.

---

**Algorithm 1** Unification via Alternating Optimization: One Single Attention Layer

---

**Require:** Training dataset of size $M$: $\{\boldsymbol{H}_i, \boldsymbol{y}_i\}_{i=1}^M$, learning rate $\eta$, training epochs $K$
**Ensure:** Updated parameters $\widehat{\boldsymbol{W}}, \widehat{\boldsymbol{E}}$ and representations $\{\hat{\boldsymbol{z}}_i\}_{i=1}^M$
  1: Initialize parameters $\boldsymbol{z}^0, \boldsymbol{E}^0, \boldsymbol{W}^0$
  2: **for** each epoch $k = 0, \dots, K - 1$ **do**      # Train for $K$ epochs with batch size $M$
  3:     **for** each sample $i = 0, \dots, M - 1$ **do**      # Local GD on $\boldsymbol{z}$ (equivalent to forward pass)
  4:         $\boldsymbol{z}_i^{k+1} = \boldsymbol{z}_i^k - \eta \nabla_{\boldsymbol{z}} F^* \left( \boldsymbol{z}_i^k, \boldsymbol{W}^k \right) = \text{Atten}(\boldsymbol{z}_i^k)$
  5:     **end for**
  6:     $\boldsymbol{W}^{k+1} = \boldsymbol{W}^k - \frac{\eta}{M} \sum_{i=1}^M \nabla_{\boldsymbol{W}} F^* \left( \boldsymbol{z}_i^{k+1}, \boldsymbol{W}^k \right)$      # Local GD on $\boldsymbol{W}$ (backpropagation)
  7:     $\boldsymbol{E}^{k+1} = \boldsymbol{E}^k - \frac{\eta}{M} \nabla_{\boldsymbol{E}} \sum_{i=1}^M \text{CE}((\boldsymbol{E}^k)^T \boldsymbol{z}_i^{k+1}, \boldsymbol{y}_i)$      # Local GD on $\boldsymbol{E}$ (backpropagation)
  8: **end for**
  9: Update $\widehat{\boldsymbol{W}} = \boldsymbol{W}^K, \widehat{\boldsymbol{E}} = \boldsymbol{E}^K$ and $\hat{\boldsymbol{z}}_i = \boldsymbol{z}_i^K$ for $i = 1, \dots, M$
10: Return $\widehat{\boldsymbol{W}}, \widehat{\boldsymbol{E}}, \{\boldsymbol{z}_i^{(K)}\}_{i=1}^M$

---

**(iii.) Residual connection and the incremental form of the update rule.** Theorem 1 shows that given parameters $\boldsymbol{W}$ and tokens $\{\boldsymbol{h}_i\}_{i=1}^N$, the forward computation of the attention layer can be modeled as one GD step minimizing the Helmholtz free energy respect to $\boldsymbol{z}$, thereby reducing the energy and driving $\boldsymbol{z}$ toward a stable position in the semantic space. In this incremental iterative update rule, the residual connection $\boldsymbol{z}^{(k)}$ serves as the current iterate (solution), the component computed by the Softmax attention provides the search direction (update), and the final output $\boldsymbol{z}^{(k+1)}$ can be viewed as the next iterate (solution).

**(iv.) Relation to Learnable Parameters in the Attention Layer.** It can be seen that the learnable $\boldsymbol{W}$ in the energy function are equivalent to $\boldsymbol{W}_Q^T \boldsymbol{W}_K$ in the attention layer, which are typically learned during training to find an appropriate semantic space for computing the free energy. Moreover, it should be noted that in practical attention layers, the learnable $\boldsymbol{W}_V$ is often not limited to form $\boldsymbol{W}_V = \eta T \boldsymbol{W}_Q^T \boldsymbol{W}_K$ but is instead more flexible, enabling the discovery of a potential better optimization path. In addition, multiple attention layers are also stacked with layer-wise parameterization, allowing for further flexibility in learning.

## 2.2 UNIFYING FORWARD AND BACKWARD VIA ALTERNATING OPTIMIZATION

In fact, by incorporating Eq (4) as a regularization term into the training objective, the model's forward inference and backward propagation during training can be unified under the perspective of alternating optimization. As a classification example, we consider a single attention layer where the input is $\boldsymbol{H} = [\boldsymbol{h}_1, \dots, \boldsymbol{h}_N] \in \mathbb{R}^{d \times N}$ (e.g., embedded image patches)[4] and $\boldsymbol{z}$ serves as a special classification token (e.g., [CLS]) to compute the final representation. The model's final output is typically projected via a projection head $\boldsymbol{E}^{d \times C}$ to obtain a logit matrix, which is then normalized by the softmax function and used to compute the cross-entropy loss, that is,

$$\text{CE}(\boldsymbol{E}^T \boldsymbol{z}, \boldsymbol{y}) = - \sum_{c=1}^C (\boldsymbol{y})_c \log \frac{e^{(\boldsymbol{E}^T \boldsymbol{z})_c}}{\sum_{u=1}^C e^{(\boldsymbol{E}^T \boldsymbol{z})_u}}, \tag{5}$$

where $C$ denotes the number of classes, $\boldsymbol{y} \in \mathbb{R}^C$ is the (soft) label vector and $(\boldsymbol{y})_c$ denotes the probability of the $c$-th class. Then $F^*$ as Eq (4) can be regarded as a regularization term on the cross-entropy loss: optimizing $\boldsymbol{z}$ in the regularization corresponds to the forward computation, while optimizing the parameters corresponds to the backward propagation that updates the model. Formally, the overall objective can be written as

$$\min_{\boldsymbol{z}, \boldsymbol{W}, \boldsymbol{E}} \text{CE} \left( \boldsymbol{E}^T \boldsymbol{z}, \boldsymbol{y} \right) + F^* \left( \boldsymbol{z}, \boldsymbol{W} \right). \tag{6}$$

The process can be described by Algorithm 1, where we train the model with $M$ samples for $K$ epochs. Within each epoch, the forward inference and backward update can be viewed as an alternating optimization process over $\boldsymbol{z}, \boldsymbol{W}$ and $\boldsymbol{E}$. More discussions can be seen in Appendix A.4.

---

[4]To avoid introducing unnecessary new notation, here we omit the update of the embedding layer.

## 3 ENERGY-BASED REFINEMENTS OF ATTENTION

In Section 2.1, we show that in our proposed energy-based framework, different combinations of the three key components will naturally give rise to corresponding attention forms, which serves as guidance for us in designing potential attention models. A natural idea then arises: if the forward pass of standard attention can be modeled as optimizing the Helmholtz free energy, can we directly obtain the final solution as the token representation (i.e., $\boldsymbol{z}^* = \operatorname{argmin}_{\boldsymbol{z}} F^*$) instead of relying on such a structure that carries out incremental updates based on local gradient descent? Unfortunately, except in certain special cases (e.g., $\boldsymbol{h}_i$ are symmetrically distributed), it is difficult to directly obtain a closed-form analytical solution for $F^*$ or its upper bound $\tilde{F}^*$. We present Lemma 2 as follows.

**Lemma 2.** *Both the Helmholtz free energy $F^*$ and its upper bound $\tilde{F}^*$ are non-convex with respect to $\boldsymbol{z}$. Assume $\|\boldsymbol{z}\| \leq \rho$ and $\|\boldsymbol{W}\boldsymbol{h}_i\| \leq \rho$ for all $i \in [N]$. The local minima of $F^*$ is attained at the boundary $\|\boldsymbol{z}\| = \rho$ or when $\boldsymbol{z} = \sum_{i=1}^{N} p_i \boldsymbol{W}\boldsymbol{h}_i$ where $p_i = \frac{1}{Z} e^{-\frac{\|\boldsymbol{z} - \boldsymbol{W}\boldsymbol{h}_i\|^2}{2T}}$ and $Z = \sum_{i=1}^{N} e^{-\frac{\|\boldsymbol{z} - \boldsymbol{W}\boldsymbol{h}_i\|^2}{2T}}$. In addition, the local minima of $\tilde{F}^*$ is attained at the boundary $\|\boldsymbol{z}\| = \rho$.*

The proof of Lemma 2 is in Appendix A.5. The core is to show the Hessian matrix of $F^*$ as

$$\nabla_{\boldsymbol{z}}^2 F^* = \underbrace{\boldsymbol{I}}_{\succeq 0} \underbrace{-\frac{1}{T} \left[ \sum_{i=1}^{N} p_i \boldsymbol{r}_i \boldsymbol{r}_i^T - (\nabla_{\boldsymbol{z}} F^*)(\nabla_{\boldsymbol{z}} F^*)^T \right]}_{\preceq 0}, \tag{7}$$

which is composed of a positive semidefinite identity matrix and a negative semidefinite term. Therefore, $F^*$ is neither convex nor concave and its local minima can only occur at the boundary or at stationary points. Similarly, the Hessian of $\tilde{F}^*$ contains only the negative semidefinite term, making it concave and ensuring that its local minima occur only on the boundary.

Although a closed-form solution is difficult to obtain directly in both cases, it is possible to obtain a better solution as the token representation by adopting more efficient GD algorithms within the energy-based framework, which in turn leads to improvements in the attention structure. As for $F^*$, the update rule derived from the first-order GD is

$$\boldsymbol{z}^{(k+1)} = \boldsymbol{z}^{(k)} - \eta \nabla_{\boldsymbol{z}^{(k)}} F^* = (1 - \eta) \boldsymbol{z}^{(k)} + \eta \bar{\boldsymbol{h}}, \tag{8}$$

where $\bar{\boldsymbol{h}} = \sum_{i=1}^{N} p_i \boldsymbol{W}\boldsymbol{h}_i$ and $p_i = \frac{1}{Z} e^{-\frac{\|\boldsymbol{z}^{(k)} - \boldsymbol{W}\boldsymbol{h}_i\|^2}{2T}}$. This can be regarded as a first-order update with momentum. A simple and straightforward idea for employing a more efficient algorithm is to use Newton's method, which leverages the second-order information from the Hessian matrix to accelerate convergence. This can be formulated as

$$\boldsymbol{z}^{(k+1)} = \boldsymbol{z}^{(k)} - \eta \left[ \nabla_{\boldsymbol{z}^{(k)}}^2 F^* \right]^{-1} \nabla_{\boldsymbol{z}^{(k)}} F^*,$$

where $\nabla_{\boldsymbol{z}^{(k)}}^2 F^*$ is the Hessian matrix at $\boldsymbol{z}^{(k)}$. In fact, using the notation $\boldsymbol{d}_i = \boldsymbol{W}\boldsymbol{h}_i - \bar{\boldsymbol{h}}$, we can rewrite the Hessian matrix in Eq (7) into a more concise form:

$$\nabla_{\boldsymbol{z}}^2 F^* = \boldsymbol{I} - \frac{1}{T} \sum_{i=1}^{N} p_i \boldsymbol{d}_i \boldsymbol{d}_i^T. \tag{9}$$

Thus the final update rule can be formed as

$$\text{Att2nd}\left(\boldsymbol{z}^{(k)}\right) = \boldsymbol{z}^{(k+1)} = \boldsymbol{z}^{(k)} - \eta \left[ \boldsymbol{I} - \frac{1}{T} \sum_{i=1}^{N} p_i \boldsymbol{d}_i \boldsymbol{d}_i^T \right]^{-1} \left( \boldsymbol{z}^{(k)} - \bar{\boldsymbol{h}} \right). \tag{10}$$

The Hessian matrix in Eq (9) is composed of a weighted covariance term and an identity matrix serving as regularization. Its inverse in Eq (10) provides a preconditioning for the first-order gradient, allowing adaptive updates along different dimensions. Corresponding to standard attention, we can also parameterize $\boldsymbol{W}$ as $\boldsymbol{W}_Q^T \boldsymbol{W}_K$ in $p_i$ while $\boldsymbol{W}$ as $\boldsymbol{W}_V$ in $\bar{\boldsymbol{h}}$ and $\boldsymbol{d}_i$ to make the model more flexible[5]. We denote this modified attention layer as $\text{Att2nd}(\boldsymbol{z})$ as it is inspired by Newton's method and uses the second-order GD information.

---

[5]The parameterization method will change in the multi-head setting. Here we mainly emphasize how to derive the Newton-inspired modification for attention and outline ideas for reducing its cost.

Recalling that the standard attention incurs a computational cost of $O(d^2 + Nd)$ per inference step with the reuse of KV caches, the cost of computing $\bar{h}$ and $d_i$ is also $O(d^2 + Nd)$. However, the inverse of the Hessian incurs a cost of $O(d^3)$ which is often impractical in practice[6]. To further reduce the cost, we approximate the inverse using its Taylor expansion, that is,

$$\left[ \boldsymbol{I} - \frac{1}{T} \sum_{i=1}^{N} p_i \boldsymbol{d}_i \boldsymbol{d}_i^T \right]^{-1} \approx \boldsymbol{I} + \frac{1}{T} \sum_{i=1}^{N} p_i \boldsymbol{d}_i \boldsymbol{d}_i^T + \frac{1}{T^2} \left( \sum_{i=1}^{N} p_i \boldsymbol{d}_i \boldsymbol{d}_i^T \right)^2 + \cdots .$$

Here, we retain only the first-order term and the approximated update rule can be reformulated as

$$\text{Att2nd1st}\left( \boldsymbol{z}^{(k)} \right) = \boldsymbol{z}^{(k+1)} = (1 - \eta)\boldsymbol{z}^{(k)} + \eta\bar{h} - \frac{\eta}{T} \sum_{i=1}^{N} p_i \boldsymbol{d}_i \boldsymbol{d}_i^T \left( \boldsymbol{z}^{(k)} - \bar{h} \right) .$$

Compared with Eq (8), the above rule adds a term that adjusts the update using weighted covariance information. By prioritizing the computation of $\boldsymbol{d}_i^T \left( \boldsymbol{z}^{(k)} - \bar{h} \right)$ to avoid matrix–vector multiplications, we can reduce the overall cost to $O(Nd + d^2)$, which is the same order as standard attention. We denote this structure as $\text{Att2nd1st}(\boldsymbol{z})$, which is inspired by Newton's method while approximating the inverse using first-order Taylor expansion. Note that our discussion so far mainly focuses on the single-head case. In the next section, we will extend to the multi-head cases and present the final modified attention along with its parameterization, following a line of ideas very similar to the discussion above.

## 4 EXTENDING THE ENERGY PRINCIPLE TO THE MULTI-HEAD CASE

Now we extend the energy principle to the multi-head case. The multi-head attention layer with $H$ heads can be formalized as

$$\text{MHA}(\boldsymbol{z}) = \boldsymbol{z} + \sum_{h=1}^{H} \sum_{i=1}^{N} \frac{e^{\boldsymbol{z}^T \boldsymbol{W}_{Q,h}^T \boldsymbol{W}_{K,h} \boldsymbol{h}_i / T}}{Z_h'} \boldsymbol{W}_{O,h} \boldsymbol{W}_{V,h} \boldsymbol{h}_i, \tag{11}$$

where $\boldsymbol{W}_{V,h}, \boldsymbol{W}_{K,h}, \boldsymbol{W}_{Q,h} \in \mathbb{R}^{d_h \times d}$ and $\boldsymbol{W}_{O,h} \in \mathbb{R}^{d \times d_h}$ are learnable parameters. In addition, we have $d_h = \frac{d}{H}$ for each head and $Z_h' = \sum_{j=1}^{N} e^{\boldsymbol{z}^T \boldsymbol{W}_{Q,h}^T \boldsymbol{W}_{K,h} \boldsymbol{h}_j / T}$ as normalizing terms. Conceptually, multi-head attention works by first projecting tokens into lower-dimensional subspaces to capture information independently and finally combining these representations back into the original $d$-dimensional space through the projection $\boldsymbol{W}_{O,h}$.

Similarly, by appropriately parameterizing $E(\boldsymbol{z}, \boldsymbol{h}_i)$, the energy arising from interactions between particles can also be modeled in $H$ low-dimensional (semantic) spaces. We denote the parameterized energy between $\boldsymbol{z}$ and $\boldsymbol{h}_i$ in the $h$-th subspace as $E_{\boldsymbol{\theta}_h}(\boldsymbol{z}, \boldsymbol{h}_i)$ where $\boldsymbol{\theta}_h$ represents the parameters. Then the average Helmholtz free energy can be defined as

$$F^* = -\frac{1}{H} \sum_{h=1}^{H} T \log Z_h = -\frac{1}{H} \sum_{h=1}^{H} T \log \sum_{i=1}^{N} e^{-\frac{E_{\boldsymbol{\theta}_h}(\boldsymbol{z}, \boldsymbol{h}_i)}{T}},$$

where $Z_h$ is the partition function for the $h$-th subspace. Here we reuse the symbols $F^*$ for the sake of notational simplicity and consistency. Next, we show that the forward computation of the multi-head attention as defined in Eq (11), can be modeled as one step GD to minimize the above average Helmholtz free energy.

**Theorem 2.** *Let the energy function $E_i = E(\boldsymbol{z}, \boldsymbol{h}_i)$ take the parameterized elastic potential form in the $h$-th subspace, that is,*

$$E_{\boldsymbol{\theta}_h}(\boldsymbol{z}, \boldsymbol{h}_i) = \frac{1}{2} \|\boldsymbol{W}_{1,h} \boldsymbol{z} - \boldsymbol{W}_{2,h} \boldsymbol{h}_i\|^2,$$

---

[6]Noting that the Hessian can be expressed as a sum of rank-1 perturbations, we can use the Sherman-Morrison-Woodbury formula to compute the inverse and the resulting cost is $O(Nd^2)$. This will provide savings when $N \ll d$, but overall, the cost is still higher than the standard attention.

*where $\boldsymbol{W}_{1,h}, \boldsymbol{W}_{2,h} \in \mathbb{R}^{d_h \times d}$ and $\boldsymbol{\theta}_h = \{\boldsymbol{W}_{1,h}, \boldsymbol{W}_{2,h}\}$ denotes the parameters. Then the average Helmholtz free energy can be formalized as*

$$F^* = -\frac{1}{H} \sum_{h=1}^{H} T \log \sum_{i=1}^{N} e^{-\frac{\|\boldsymbol{W}_{1,h}\boldsymbol{z} - \boldsymbol{W}_{2,h}\boldsymbol{h}_i\|^2}{2T}}.$$

*Assuming that $\|\boldsymbol{W}_{1,h}\boldsymbol{z}\| = \|\boldsymbol{W}_{2,h}\boldsymbol{h}_i\| = \rho$ for all $i \in [N], h \in [H]$, the forward inference of the multi-head attention defined in Eq (11) can be modeled as one gradient descent step for minimizing $F^*$ with the learning rate $\eta$ when setting $\boldsymbol{W}_{Q,h}^T \boldsymbol{W}_{K,h} = \boldsymbol{W}_{1,h}^T \boldsymbol{W}_{2,h}$ and $\boldsymbol{W}_{O,h} \boldsymbol{W}_{V,h} = \frac{\eta T}{H} \boldsymbol{W}_{1,h}^T \boldsymbol{W}_{2,h}$ for all $h \in [H]$.*

The proof can be seen in Appendix A.6. It can be noticed that the energy function here still takes the form of elastic potential. However, unlike the original approach that only applies $\boldsymbol{W}$ to $\boldsymbol{h}$, here we introduce $\boldsymbol{W}_{1,h}, \boldsymbol{W}_{2,h}$ to embed both $\boldsymbol{z}$ and $\boldsymbol{h}_i$ for the $h$-th space, allowing the energy computation to be carried out independently across each semantic subspace. In the multi-head setting, we still cannot obtain a closed-form convergence solution (see Lemma 5 in Appendix A.7).

As in the single-head case, we can also extend the Newton's method-inspired attention modification to the multi-head setting. We denote the Helmholtz free energy in the $h$-th subspace as $F_h^* = -T \log \sum_{i=1}^{N} Z_h$ and then $F^* = \frac{1}{H} F_h^*$. Instead of applying Newton's method directly to $F^*$, we apply it independently to each subspace $F_h^*$, which can be formalized as

$$\boldsymbol{z}^{(k+1)} = \boldsymbol{z}^{(k)} - \frac{\eta}{H} \sum_{h=1}^{H} \left[ \nabla_{\boldsymbol{z}^{(k)}}^2 F_h^* \right]^{-1} \nabla_{\boldsymbol{z}^{(k)}} F_h^*$$

Considering the analogous roles of $\boldsymbol{W}_{1,h}^T \boldsymbol{W}_{2,h}$ and $\boldsymbol{W}_{Q,h}^T \boldsymbol{W}_{K,h}$ in Theorem 2, we use the notation $\boldsymbol{q}_h = \boldsymbol{W}_{1,h} \boldsymbol{z}$, $\boldsymbol{k}_{i,h} = \boldsymbol{W}_{2,h} \boldsymbol{h}_i$ and $\bar{\boldsymbol{k}}_h = \sum_{i=1}^{N} p_{i,h} \boldsymbol{W}_{2,h} \boldsymbol{h}_i$ where $p_{i,h} = \frac{1}{Z_h} e^{-\frac{\|\boldsymbol{W}_{1,h}\boldsymbol{z} - \boldsymbol{W}_{2,h}\boldsymbol{h}_i\|^2}{2T}}$. Then the Hessian matrix of $F_h^*$ is

$$\nabla_{\boldsymbol{z}}^2 F_h^* = \boldsymbol{W}_{1,h}^T \left[ \boldsymbol{I} - \frac{1}{T} \sum_{i=1}^{N} p_{i,h} \left( \boldsymbol{k}_{i,h} - \bar{\boldsymbol{k}}_h \right) \left( \boldsymbol{k}_{i,h} - \bar{\boldsymbol{k}}_h \right)^T \right] \boldsymbol{W}_{1,h}.$$

Note that due to $\boldsymbol{W}_{1,h} \in \mathbb{R}^{d_h \times d}$, the Hessian matrix $\nabla_{\boldsymbol{z}}^2 F_h^* \in \mathbb{R}^{d \times d}$ is non-invertible. Therefore, we need to employ the range-space approach[7] to compute the inverse. Furthermore, to reduce the computational cost, we also approximate the inverse of the intermediate matrix using a first-order Taylor expansion. Finally, by parameterize $\boldsymbol{W}_{1,h}, \boldsymbol{W}_{2,h}$ as $\boldsymbol{W}_{Q,h}, \boldsymbol{W}_{K,h}$, the Att2nd1st$(\boldsymbol{z})$ can be extended as

$$\text{MHA2nd1st}(\boldsymbol{z}) = \boldsymbol{z} + \frac{\eta}{H} \sum_{h=1}^{H} \boldsymbol{W}_{Q,h}^T \left[ (\boldsymbol{q}_h - \bar{\boldsymbol{k}}_h) + \boldsymbol{b}_h \right],$$

$$\boldsymbol{b}_h = \frac{1}{T} \left( \boldsymbol{W}_{Q,h} \boldsymbol{W}_{Q,h}^T \right)^{-1} \sum_{i=1}^{N} p_{i,h} \boldsymbol{d}_{i,h} \left[ \boldsymbol{d}_{i,h}^T \boldsymbol{W}_{Q,h} \boldsymbol{W}_{Q,h}^T (\boldsymbol{q}_h - \bar{\boldsymbol{k}}_h) \right].$$

where $\boldsymbol{d}_{i,h} = \boldsymbol{k}_{i,h} - \bar{\boldsymbol{k}}_h$. We can see that the vector $\boldsymbol{b}_h$ acts as a bias term, adjusting the update using variance information in the subspace. In practice, we introduce new parameters $\boldsymbol{W}_O \in \mathbb{R}^{d \times d_h}$ to replace $\frac{\eta}{H} \boldsymbol{W}_{Q,h}^T$ to make the model more flexible. Moreover, to maintain stability, we set the temperature $T$ in the attention score $p_{i,h}$ as a head-wise learnable parameter with initialization as $d_h$ and the temperature in $\boldsymbol{b}_h$ is treated in the same way. Compared with standard attention, the final structure keeps $\boldsymbol{W}_Q, \boldsymbol{W}_V, \boldsymbol{W}_O$ while removing the value mapping $\boldsymbol{W}_V$, thereby reducing the number of parameters by one quarter. Meanwhile, the total cost for H heads is $O(Nd + d^2)$, sharing the same asymptotic complexity as standard attention despite a larger constant factor. More details can be seen in Appendix A.8.

---

[7]Here we use $\left( \boldsymbol{W}^T \boldsymbol{C} \boldsymbol{W} \right)^\dagger = \boldsymbol{W}^T \left( \boldsymbol{W} \boldsymbol{W}^T \right)^{-1} \boldsymbol{C}^{-1} \boldsymbol{W}$ when $\boldsymbol{W} \in \mathbb{R}^{m \times n}$ and $m < n$.

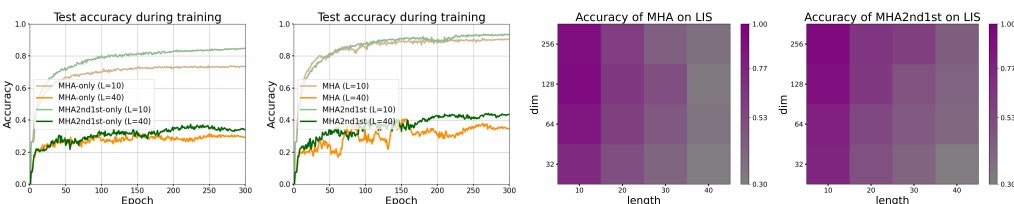

Figure 1: **Left Part:** Test accuracy during training when the task length $L = 10/40$ and model size $d = 256$: attention-only layers (leftmost) and alternating attention–FFN layers (center left). **Right Part:** Test accuracy of MHA and MHA2nd1st across different task lengths and model sizes.

## 5 EXPERIMENTS

Following the setup of Feng et al. (2024); Yang et al. (2024a), we evaluate the capability of the proposed attention structure in solving a classical dynamic programming (DP) problem—the Longest Increasing Subsequence (LIS) task. Given the a sequence $s \in \mathbb{N}^L$ of length $L$, a sequence $\tilde{s}$ is the subsequence of $s$ if there exists an index set $1 \leq i_1 < i_2 < \cdots < i_{|\tilde{s}|} \leq n$ such that $\tilde{s}_k = s_{i_k}$ for all $k \in [|\tilde{s}|]$. A subsequence $\tilde{s}$ is called increasing if it satisfies that $\tilde{s}_1 < \tilde{s}_2 < \cdots < \tilde{s}_{|\tilde{s}|}$. The goal of the LIS task is to predict the length of the longest increasing subsequence.

In our experiments[8], we control the scale of the problem (i.e., the sequence length $L$) and the model size (i.e., the model dimension $d$) to investigate the model's ability to solve the task. We use the standard Transformer model (Vaswani et al., 2017) and replaced the original multi-head attention layer (MHA) with the proposed MHA2nd1st. As mentioned in Section 4, since $\boldsymbol{W}_V$ is removed, the replaced attention layer reduces the number of parameters by 1/4. All models were trained from scratch using a draft model. During training, the model is optimized using cross-entropy loss on the answer tokens, while a greedy decoding strategy is employed during testing. More experimental and results details can be seen in AppendixA.9.

First, to more directly compare the original attention with our proposed one, we remove the Feed-Forward Network (FFN) layers from the Transformer and retain only the attention layers, labeled as MHA-only and MHA2nd1st-only respectively. We present the test accuracy during training in the leftmost panel of Figure 1. When the problem size is small, MHA2nd1st-only improves more rapidly and achieves higher accuracy. As the problem size increases, the accuracy of both models declines while MHA2nd1st-only still maintains the advantage. Furthermore, we retain the original FFN layers in the center-left part of Figure 1. It can be seen that adding FFN layers improves performance for both models under the same problem size, yet MHA2nd1st still outperforms the original MHA. In the right part of Figure 1, we further present the performance of the two models under different task lengths and model sizes. It can be seen that MHA2nd1st overall outperforms MHA, especially when the problem size is large. These results provide preliminary support that the modified attention structure derived from the energy-based framework has the potential to use fewer parameters to achieve performance that is comparable to or exceeds that of the original MHA.

## 6 CONCLUSION

In this work, we revisit the energy principle to understand attention-based Transformers. We propose an energy-based framework whose key components include the global energy $F^*$, the energy function $E_i$ and the form of gradient descent to explain both attention structures. Within this framework, the forward inference of standard attention can be seen as a special case where $F^*$ corresponds to the Helmholtz free energy, $E_i$ takes the form of an elastic potential and first-order gradient descent is employed. Based on this connection, we note that the forward pass and parameter backpropagation can be unified under an alternating optimization perspective. Furthermore, inspired by Newton's method, we extend the original first-order GD-based standard attention to a second-order form, which leverages covariance information to adjust the updates. Our experimental results provide preliminary support for the potential of our proposed attention structure.

---

[8]Code is anonymized at `https://anonymous.4open.science/r/energy-attn-A23C`

## ETHICS STATEMENT

The authors have read and adhered to the ICLR Code of Ethics. This work is primarily theoretical and aims to understand attention based on the proposed energy-based framework. Our research does not involve human subjects or the collection of new sensitive data. There may be some potential societal consequences of our work, none of which we feel must be specifically highlighted here.

## REPRODUCIBILITY STATEMENT

To ensure the reproducibility of our findings, we have made our resources publicly available. For the theoretical contributions, we provide complete mathematical derivations and proofs for all theorems and lemmas in Appendix A. For empirical results, all code used for the experiments presented in this paper can be accessed through the anonymous GitHub repository linked in the Section 5. A comprehensive description of the experimental setup is detailed in Appendix A.9. We believe these resources provide the necessary details for the research community to verify our claims and build upon our work.

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

## USE OF LARGE LANGUAGE MODELS

In line with the ICLR policy, we disclose that Large Language Models (LLMs) were used as a general-purpose writing assistant during the preparation of this manuscript. The primary role of LLMs was to aid in polishing the text, which included improving grammar, refining sentence structure for clarity, and checking for stylistic consistency.

## A  APPENDIX

### A.1  DISCUSSIONS ON RELATED WORK AND FUTURE DIRECTION

In this part, we discuss the related work and potential future directions in more detailed discussion.

**Energy principle and Transformers:** The concept of energy has long been used in deep neural networks (Hopfield, 1982; 1984; Ackley et al., 1985; Krotov & Hopfield, 2016; LeCun et al., 2006; LeCun, 2022). Previous work has also linked energy to the attention mechanism in Transformers and the studies most relevant to ours are likely those by Ramsauer et al. (2020) and Hoover et al. (2023). Ramsauer et al. (2020) proposed a new energy function for continuous-state Hopfield networks and pointed out that this Hopfield update rule corresponds to the attention mechanism in the Transformer. Hoover et al. (2023) also proposed the Multi-Head Energy Attention, whose dynamic evolution includes the computational process of standard attention. In this work, we revisit the energy perspective to interpret the attention mechanism. However, unlike previous works, we extend the interpretation of standard attention into a more general framework, which consists of three key components: the Global Energy $F^*$, the Energy function $E_i$, and the Gradient Descent (GD) form. We illustrate that standard attention is only a special case within this framework. For instance, by altering the form of the energy, we can derive the formulation of linear attention (see Appendix A.3)); and by extending the GD form from first-order to second-order gradient descent, we arrive at the proposed MHA2nd1st. Furthermore, we note that Gladstone et al. (2025) employ energy-based methods to train Transformers and their focus is more related to training paradigms. We believe this is orthogonal to our work.

**Unrolled Optimization and Model Architecture:** Understanding and designing model architectures from the perspective of unrolled optimization is a currently active area of research (Gregor & LeCun, 2010; Tolooshams & Ba, 2021; Chan et al., 2022). Previous works have designed and interpreted Transformer-like structures from various viewpoints, including sparse rate distortion (Yu et al., 2024b;a), denoising (Wang et al., 2025b), information bottleneck (Zhou et al., 2022), multinomial regression (Actor et al., 2025), etc. Unlike previous work, our approach starts from the concept of energy to interpret the standard attention mechanism, and show that new structure can be designed based on the proposed energy framework. We also note that some other studies focus more on leveraging an optimization perspective to guide the design of more efficient model architectures (e.g., those with linear complexity with respect to sequence length) (von Oswald et al., 2025; Yang et al., 2024b; Wang et al., 2025a). We believe that the energy-based framework holds potential for designing more efficient attention structures in the future, possibly through the development of novel energy functions or GD forms. Additionally, our proposed attention mechanism is primarily inspired by Newton's method. In fact, numerous first-order optimization algorithms (e.g., Adam (Kingma, 2014)) could also inspire further improvements to existing attention mechanisms. Although we employed a first-order Taylor approximation to reduce the computational cost of the Newton-inspired attention to the same order as standard attention, it still carries a larger constant factor. We believe that other techniques, such as random feature methods (Yu et al., 2016; Choromanski et al., 2020), could be used to approximate the relevant operations, potentially achieving even lower computational costs than standard attention.

**Test-time Scaling and Loop Transformers:** Test-time scaling is a favored pathway to boost model inference (Zhang et al., 2025; Snell et al., 2024; Muennighoff et al., 2025). Among these methods, Loop Transformers output token representations through parameter-shared recurrent computations and existing research demonstrates that this recurrent structure offers advantages in terms of performance gains and capability generalization (Geiping et al., 2025; Fan et al., 2024; Yang et al., 2023; Yu et al., 2025). As mentioned in Appendix A.4, unlike stacking attention layers with distinct parameters, using parameter-shared recurrent computation aligns more closely with optimizing the same energy function within a relatively stable semantic space. Therefore, we believe a promising

direction is to apply the attention mechanism induced by higher-order GD forms within Loop Transformers to enable more "efficient" representation learning. Additionally, enhancing the capacity of attention in a parameter-free manner, using approaches like MHA2nd1st, could represent another viable path for test-time scaling. Concurrently, increasing the computational overhead of attention without introducing extra parameters, following a paradigm like MHA2nd1st, may represent another potential path for test-time scaling.

## A.2 PROOF OF LEMMA 1

**Lemma 3** (Helmholtz free energy). *Define the partition function as $Z = \sum_{i=1}^{N} e^{-E_i/T}$. The system's free energy defined by Eq (2) attains its minimum value*

$$F^* = -T \log Z = -T \sum_{i=1}^{N} e^{-E_i/T}, \tag{12}$$

*when $p_i$ satisfies the Boltzmann distribution, i.e., $p_i = \frac{e^{-E_i/T}}{Z}$.*

*Proof.* The problem can be formed as

$$\min_{p_1, p_2, \dots, p_N} F = \sum_{i=1}^{N} p_i E_i + T \sum_{i=1}^{N} p_i \log p_i \quad s.t. \quad \sum_{i=1}^{N} p_i = 1.$$

We can use a Lagrange multiplier $\alpha$ for the equality constraint:

$$\mathcal{L} = \sum_{i=1}^{N} p_i E_i + T \sum_{i=1}^{N} p_i \log p_i - \alpha \left( \sum_{i=1}^{N} p_i - 1 \right).$$

Then, we can get the stationarity w.r.t. $p_i$ as:

$$\frac{\partial \mathcal{L}}{\partial p_i} = E_i + T \left( \log p_i + 1 \right) - \alpha = 0.$$

Thus, we have

$$p_i = e^{\alpha/T - 1} e^{-E_i/T} \Rightarrow p_i \propto e^{-E_i/T},$$

where $\alpha$ should scale $e^{-E_i/T}$ so that the constraint $\sum_{i=1}^{N} p_i = 1$ is satisfied. Therefore, we obtain $p_i = \frac{e^{-E_i/T}}{Z}$ where $Z = \sum_{i=1}^{N} e^{-E_i/T}$ is the partition function. Then, we have

$$F^* = \sum_{i=1}^{N} p_i E_i + T \sum_{i=1}^{N} p_i \log \frac{e^{-E_i/T}}{Z} = -T \log Z.$$

Finally, the minimizer is unique because $F$ is convex on the simplex. Thus, we complete our proof. $\square$

## A.3 LINEAR ATTENTION WITHIN THE ENERGY-BASED FRAMEWORK

For a given input $\boldsymbol{z} \in \mathbb{R}^d$, we assume that the set of tokens interacting with it is $\{\boldsymbol{h}_i\}_{i=1}^{N} \in \mathbb{R}^{d \times N}$. The linear attention can be formalized as

$$\text{LinearAtten}(\boldsymbol{z}) = \boldsymbol{z} + \sum_{i=1}^{N} \left( \boldsymbol{z}^T \boldsymbol{W}_Q^T \boldsymbol{W}_K \boldsymbol{h}_i \right) \boldsymbol{W}_V \boldsymbol{h}_i, \tag{13}$$

where $\boldsymbol{W}_Q, \boldsymbol{W}_K, \boldsymbol{W}_V \in \mathbb{R}^{d \times d}$ are learnable parameters for query, key and value projection. Compared to standard attention, it eliminates the need for the softmax operation on attention scores. The following theorem shows that when we alter the forms of the global energy $F^*$ and the energy function $E_i$ within the energy framework in Table 1, the forward inference of linear attention can still be viewed as minimizing $F^*$ using first-order gradient descent.

**Theorem 3.** *Let the energy function $E_i = E(\boldsymbol{z}, \boldsymbol{h}_i)$ take the parameterized inner product form, that is,*

$$E_{\boldsymbol{W}}(\boldsymbol{z}, \boldsymbol{h}_i) = -\boldsymbol{z}^T \boldsymbol{W} \boldsymbol{h}_i,$$

*where $\boldsymbol{W} \in \mathbb{R}^{d \times d}$ is the learnable parameter. Let the Global Energy $F^*$ take the form of a sum of squares, which can be formalized as*

$$F^* = -\frac{T}{2} \sum_{i=1}^{N} E_i^2 = -\frac{T}{2} \left( \boldsymbol{z}^T \boldsymbol{W} \boldsymbol{h}_i \right)^2. \tag{14}$$

*Then the forward inference of linear attention in Eq (13) can be modeled as one gradient descent step for minimizing $F^*$ with the learning rate $\eta$ when setting $\boldsymbol{W}_Q^T \boldsymbol{W}_K = \boldsymbol{W}$ and $\boldsymbol{W}_V = \eta T \boldsymbol{W}$.*

*Proof.* We can take the derivative of $F^*$ with respect to $z$ to obtain

$$\nabla_{\boldsymbol{z}} F^* = -\sum_{i}^{N} \nabla_{\boldsymbol{z}} \left( \boldsymbol{z}^T \boldsymbol{W} \boldsymbol{h}_i \right)^2 = -\sum_{i=1}^{N} \left( \boldsymbol{z}^T \boldsymbol{W} \boldsymbol{h}_i \right) \boldsymbol{W} \boldsymbol{h}_i.$$

Then, given an initial value $\boldsymbol{z}^{(0)}$, we can apply gradient descent to minimize the objective $F^*$. Suppose the learning rate is $\eta$, the iteration is given by

$$\boldsymbol{z}^{(k+1)} = \boldsymbol{z}^{(k)} - \eta \nabla_{\boldsymbol{z}^{(k)}} F^* = \boldsymbol{z}^{(k)} + \sum_{i=1}^{N} \left( (\boldsymbol{z}^{(k)})^T \boldsymbol{W} \boldsymbol{h}_i \right) \eta T \boldsymbol{W} \boldsymbol{h}_i.$$

By comparing with Eq (13), we can rewrite the learnable $\boldsymbol{W}$ as $\boldsymbol{W} = \boldsymbol{W}_Q^T \boldsymbol{W}_K$ and further set $\eta T \boldsymbol{W} = \boldsymbol{W}_V$. Then, we will have

$$\boldsymbol{z}^{(k+1)} = \text{LinearAtten}(\boldsymbol{z}^{(k)}) = \boldsymbol{z}^{(k)} + \sum_{i=1}^{N} \left( (\boldsymbol{z}^{(k)})^T \boldsymbol{W}_Q^T \boldsymbol{W}_K \boldsymbol{h}_i \right) \boldsymbol{W}_V \boldsymbol{h}_i,$$

which has the same form as the linear attention layer in Eq (13). Thus, we complete our proof. $\square$

### A.4 MORE DISCUSSIONS ON LOOP TRANSFORMERS

While attention layers are commonly stacked with varying parameters across layers, Loop Transformers usually share parameters across iterations, helping preserve a relatively stable semantic space. In this case, the forward loop computation can be modeled as alternately updating $F^*(\boldsymbol{z}_i, \boldsymbol{H}, \boldsymbol{W})$ with respect to $\boldsymbol{z}_i$ at each position, given the shared $\boldsymbol{W}$ and the corresponding $\boldsymbol{H}$ composed of attended set. Taking causal attention as an example, for the $i$-th position, the attended set typically consists of the preceding tokens $\boldsymbol{H}_{\leq i} = [\boldsymbol{h}_1, \ldots, \boldsymbol{h}_i]$. Then the global objective is

$$\min_{\boldsymbol{Z}, \boldsymbol{H}} \sum_{i=1}^{N} F^* \left( \boldsymbol{z}_i, \boldsymbol{H}_{\leq i}, \boldsymbol{W} \right) \quad s.t. \ \boldsymbol{Z} = \boldsymbol{H}, \tag{15}$$

where $\boldsymbol{Z} = [\boldsymbol{z}_1, \ldots, \boldsymbol{z}_N] \in \mathbb{R}^{d \times N}$. The constraint ensures that after each iteration, the tokens used in attended sets are aligned with the newly updated $\boldsymbol{Z}$. The iteration starts with the initialization $\boldsymbol{z}_i^0 = \boldsymbol{h}_i^0 = \boldsymbol{h}_i$. The forward computation of a single-layer Loop Transformer with $K$ iterations can be equivalently viewed as performing $K$ steps of gradient descent on each $\boldsymbol{z}$, which can be described by Alogrithm 2

**Unifying forward inference and backpropagation via alternating optimization.** In fact, by incorporating Eq (15) as a regularization term into the training objective, the model's forward inference and backward propagation can be unified under the perspective of alternating optimization. For example, in autoregressive training, the model's final output representations $\boldsymbol{Z}$ are typically projected onto the vocabulary to obtain a logit matrix, which is then normalized by the softmax function and used to compute the cross-entropy loss, that is,

$$\mathcal{L}(\boldsymbol{E}^T \boldsymbol{Z}, \boldsymbol{Y}) = -\sum_{i=1}^{N} \sum_{v=1}^{V} (\boldsymbol{y}_i)_v \log \frac{e^{(\boldsymbol{E}^T \boldsymbol{z}_i)_v}}{\sum_{u=1}^{V} e^{(\boldsymbol{E}^T \boldsymbol{z}_i)_u}}, \tag{16}$$

---

**Algorithm 2** The Forward Inference of One-Layer Loop Transformer

---

**Require:** Learned $\boldsymbol{W}$, Tokens $\{\boldsymbol{h}_i\}_{i=1}^N$, temperature $T$, learning rate $\eta$
**Ensure:** Updated representation $\{\boldsymbol{z}_i^K\}_{i=1}^N$
  1: Initialize $\boldsymbol{z}_i^0 = \boldsymbol{h}_i^0 = \boldsymbol{h}_i$ for $i = 1, \dots, N$
  2: **for** each iteration $k = 0, \dots, K - 1$ **do**     # $K$ iterations of Loop Transformer
  3:      **for** each position $i = 1, \dots, N$ **do**     # Local GD on each $\boldsymbol{z}$ (equivalent to forward pass)
  4:         Update $\boldsymbol{z}_i^{k+1} = \boldsymbol{z}_i^k - \eta \nabla_{\boldsymbol{z}_i^k} F^* \left( \boldsymbol{z}_i^k, \boldsymbol{H}_{\leq i}^k; \boldsymbol{W} \right) = \text{Atten}(\boldsymbol{z}_i^k)$
  5:      **end for**
  6:      Update $\boldsymbol{h}_i^{k+1} = \boldsymbol{z}_i^{k+1}$ for $i = 1, \dots, N$
  7: **end for**
  8: Return $\{\boldsymbol{z}_i^K\}_{i=1}^N$

---

where $V$ is the vocabulary size, $\boldsymbol{E} \in \mathbb{R}^{d \times V}$ is the final projection matrix and $\boldsymbol{Y} = [\boldsymbol{y}_1, \dots, \boldsymbol{y}_N] \in \mathbb{R}^{V \times N}$ is the label matrix often composed of $N$ one-hot vectors. We also call $\boldsymbol{E}^T \boldsymbol{Z} \in \mathbb{R}^{V \times N}$ as the unnormalized logit matrix. Eq (15) can be regarded as a regularization term on the autoregressive loss: optimizing the representations $\boldsymbol{Z}$ in the regularization corresponds to the forward computation, while optimizing the parameters corresponds to the backward propagation that updates the model. Formally, the overall objective can be written as

$$\min_{\boldsymbol{Z}, \boldsymbol{H}, \boldsymbol{W}, \boldsymbol{E}} \mathcal{L} \left( \boldsymbol{E}^T \boldsymbol{Z}, \boldsymbol{Y} \right) + \sum_{i=1}^N F^* \left( \boldsymbol{z}_i, \boldsymbol{H}_{\leq i}; \boldsymbol{W} \right), \quad s.t. \ \boldsymbol{Z} = \boldsymbol{H}, \tag{17}$$

where $\mathcal{L}$ is the cross-entropy loss as Eq 16. A single forward inference and backward update can be viewed as an alternating optimization process over $\boldsymbol{Z}$ (or $\boldsymbol{H}$), $\boldsymbol{W}$, and $\boldsymbol{E}$, which can be described by Algorithm 3. In this way, the forward and backward processes can be unified as performing local GD on the regularized training loss, where the form of the regularization term is determined by the model architecture.

---

**Algorithm 3** Unification via Alternating Optimization: One-Layer Loop Transformer

---

**Require:** Tokens $\{\boldsymbol{h}_i\}_{i=1}^N$, temperature $T$, learning rate $\eta$
**Ensure:** Updated representation $\{\boldsymbol{z}_i^K\}_{i=1}^N$, updated parameters $\widehat{\boldsymbol{W}}, \widehat{\boldsymbol{E}}$
  1: Initialize parameters $\boldsymbol{E}, \boldsymbol{W}$ and $\boldsymbol{z}_i^0 = \boldsymbol{h}_i^0 = \boldsymbol{h}_i$ for $i = 1, \dots, N$
  2: **for** each iteration $k = 0, \dots, K - 1$ **do**     # $K$ iterations of Loop Transformer
  3:      **for** each position $i = 1, \dots, N$ **do**     # Local GD on $\boldsymbol{z}$ (equivalent to forward pass)
  4:         Update $\boldsymbol{z}_i^{k+1} = \boldsymbol{z}_i^k - \eta_k \nabla_{\boldsymbol{z}_i^k} F^* \left( \boldsymbol{z}_i^k, \boldsymbol{H}_{\leq i}^k, \boldsymbol{W} \right) = \text{Atten}(\boldsymbol{z}_i^k)$
  5:      **end for**
  6:      Update $\boldsymbol{h}_i^{k+1} = \boldsymbol{z}_i^{k+1}$ for $i = 1, \dots, N$
  7: **end for**
  8: Update $\widehat{\boldsymbol{W}} = \boldsymbol{W} - \eta \nabla_{\boldsymbol{W}} F^* \left( \boldsymbol{z}_i^k, \boldsymbol{H}_{\leq i}^k; \boldsymbol{W} \right)$     # Local GD on $\boldsymbol{W}$ (backpropagation)
  9: Update $\widehat{\boldsymbol{E}} = \boldsymbol{E} - \eta \nabla_{\boldsymbol{E}} \mathcal{L}(\boldsymbol{E}^T \boldsymbol{Z}^K, \boldsymbol{Y})$     # Local GD on $\boldsymbol{E}$ (backpropagation)
10: Return $\widehat{\boldsymbol{W}}, \widehat{\boldsymbol{E}}, \{\boldsymbol{z}_i^K\}_{i=1}^N$

---

## A.5    Proof of Lemma 2

**Lemma 4.** *Both the Helmholtz free energy $F^*$ with respect to $\boldsymbol{z}$ and its upper bound $\tilde{F}^*$ are non-convex. Assume $\|\boldsymbol{z}\| \leq \rho$ and $\|\boldsymbol{W}\boldsymbol{h}_i\| \leq \rho$ for all $i \in [N]$. The local minima of $F^*$ is attained at the boundary $\|\boldsymbol{z}\| = \rho$ or when $\boldsymbol{z} = \sum_{i=1}^N p_i \boldsymbol{W}\boldsymbol{h}_i$ where $p_i = \frac{1}{Z} e^{-\frac{\|\boldsymbol{z} - \boldsymbol{W}\boldsymbol{h}_i\|^2}{2T}}$ and $Z = \sum_{i=1}^N e^{-\frac{\|\boldsymbol{z} - \boldsymbol{W}\boldsymbol{h}_i\|^2}{2T}}$. In addition, the local minima of $\tilde{F}^*$ is attained at the boundary $\|\boldsymbol{z}\| = \rho$.*

*Proof.* Recalling that $F^* = -T \log \sum_{i=1}^{N} e^{-\frac{\|\boldsymbol{z} - \boldsymbol{W} \boldsymbol{h}_i\|^2}{2T}}$. We can compute the derivative of $F^*$ with respect to $\boldsymbol{z}$ as

$$\nabla_{\boldsymbol{z}} F^* = -T \nabla_{\boldsymbol{z}} \log \sum_{i=1}^{N} e^{-\frac{\|\boldsymbol{z} - \boldsymbol{W} \boldsymbol{h}_i\|^2}{2T}} = \sum_{i=1}^{N} p_i (\boldsymbol{z} - \boldsymbol{W} \boldsymbol{h}_i),$$

where $p_i = \frac{1}{Z} e^{-\frac{\|\boldsymbol{z} - \boldsymbol{W} \boldsymbol{h}_i\|^2}{2T}}$ and $Z = \sum_{i=1}^{N} e^{-\frac{\|\boldsymbol{z} - \boldsymbol{W} \boldsymbol{h}_i\|^2}{2T}}$. For notational simplicity, we denote $\boldsymbol{r}_i = \boldsymbol{z} - \boldsymbol{W} \boldsymbol{h}_i$. To compute the Hessian matrix, we first calculate

$$\nabla_{\boldsymbol{z}} p_i = \nabla_{\boldsymbol{z}} \frac{e^{-\frac{\|\boldsymbol{r}_i\|^2}{2T}}}{Z} = \frac{-\frac{1}{T} \boldsymbol{r}_i e^{-\frac{\|\boldsymbol{r}_i\|^2}{2T}} Z - e^{-\frac{\|\boldsymbol{r}_i\|^2}{2T}} \sum_{j=1}^{N} e^{-\frac{\|\boldsymbol{r}_j\|^2}{2T}} \left( -\frac{\boldsymbol{r}_j}{T} \right)}{Z^2}$$

$$= -\frac{1}{T} p_i \boldsymbol{r}_i + \frac{1}{T} p_i \sum_{j=1}^{N} p_j \boldsymbol{r}_j$$

Therefore, the Hessian matrix of $F^*$ with respect to $\boldsymbol{z}$ is

$$\nabla_{\boldsymbol{z}}^2 F^* = \sum_{i=1}^{N} \boldsymbol{r}_i \left( -\frac{1}{T} p_i \boldsymbol{r}_i^T + \frac{1}{T} p_i \sum_{j=1}^{N} p_j \boldsymbol{r}_j^T \right) + \boldsymbol{I} = \boldsymbol{I} - \frac{1}{T} \sum_{i=1}^{N} p_i \boldsymbol{r}_i \boldsymbol{r}_i^T + \frac{1}{T} \sum_{i=1}^{N} p_i \boldsymbol{r}_i \sum_{j=1}^{N} p_j \boldsymbol{r}_j^T$$

$$= \boldsymbol{I} - \frac{1}{T} \left[ \sum_{i=1}^{N} p_i \boldsymbol{r}_i \boldsymbol{r}_i^T - (\nabla_{\boldsymbol{z}} F^*) (\nabla_{\boldsymbol{z}} F^*)^T \right].$$

Furthermore, for any $\boldsymbol{v} \in \mathbb{R}^d$, we have

$$\boldsymbol{v}^T \nabla_{\boldsymbol{z}}^2 F^* \boldsymbol{v} = \|\boldsymbol{v}\|^2 - \frac{1}{T} \left[ \sum_{i=1}^{N} p_i \boldsymbol{v}_i^T \boldsymbol{r}_i \boldsymbol{r}_i^T \boldsymbol{v}_i - \left( \boldsymbol{v}^T \nabla_{\boldsymbol{z}} F^* \right) \left( \boldsymbol{v}^T F^* \right)^T \right] \tag{18}$$

Let $X_i = \boldsymbol{r}_i^T \boldsymbol{v}$ and define a random variable $X$ such that $P(X = X_i) = p_i$. Then for the second term in Eq (18), we have

$$-\frac{1}{T} \left[ \sum_{i=1}^{N} p_i \|\boldsymbol{r}_i^T \boldsymbol{v}\|^2 - \left( \sum_{i=1}^{N} p_i \boldsymbol{r}_i^T \boldsymbol{v} \right)^2 \right] = -\frac{1}{T} \left[ \mathbb{E} \left( X_i^2 \right) - \mathbb{E}^2 \left( X_i \right) \right] = -\frac{1}{T} \mathrm{Var}(X) \le 0.$$

Considering that the identity matrix is positive semi-definite, we obtain

$$\nabla_{\boldsymbol{z}}^2 F^* = \underbrace{\boldsymbol{I}}_{\succeq 0} \underbrace{-\frac{1}{T} \left[ \sum_{i=1}^{N} p_i \boldsymbol{r}_i \boldsymbol{r}_i^T - (\nabla_{\boldsymbol{z}} F^*)(\nabla_{\boldsymbol{z}} F^*)^T \right]}_{\preceq 0}.$$

Therefore, we obtain that $F^*$ is neither convex nor concave and when $\|\boldsymbol{z}\| \le \rho$, its local minima can only be attained at the boundary $\|\boldsymbol{z}\| = \rho$ or at interior points where $\nabla_{\boldsymbol{z}} F^* = 0$, that is, $\boldsymbol{z} = \sum_{i=1}^{N} p_i \boldsymbol{W} \boldsymbol{h}_i$.

Similarly, we can obtain the Hessian matrix of $\tilde{F}^*$ as

$$\nabla_{\boldsymbol{z}}^2 \tilde{F}^* = -\frac{1}{T} \left[ \sum_{i=1}^{N} p_i (\boldsymbol{W} \boldsymbol{h}_i)(\boldsymbol{W} \boldsymbol{h}_i)^T - (\nabla_{\boldsymbol{z}} \tilde{F}^*)(\nabla_{\boldsymbol{z}} \tilde{F}^*)^T \right] \preceq 0,$$

where $p_i = \frac{e^{\boldsymbol{z}^T \boldsymbol{W} \boldsymbol{h}_i / T}}{Z}$ and $Z = \sum_{i=1}^{N} e^{\frac{\boldsymbol{z}^T \boldsymbol{W} \boldsymbol{h}_i}{T}}$. Therefore, we can get that $\tilde{F}^*$ is concave and when $\|\boldsymbol{z}\| \le \rho$, its local minima can only be attained at the boundary $\|\boldsymbol{z}\| = \rho$. $\square$

### A.6 Proof of Theorem 2

**Theorem 4.** *Let the energy function $E_i = E(\boldsymbol{z}, \boldsymbol{h}_i)$ take the parameterized elastic potential form in the $h$-th subspace, that is,*

$$E_{\boldsymbol{\theta}_h}(\boldsymbol{z}, \boldsymbol{h}_i) = \frac{1}{2} \|\boldsymbol{W}_{1,h} \boldsymbol{z} - \boldsymbol{W}_{2,h} \boldsymbol{h}_i\|^2,$$

*where $\boldsymbol{W}_{1,h}, \boldsymbol{W}_{2,h} \in \mathbb{R}^{d_h \times d}$ and $\boldsymbol{\theta}_h = \{\boldsymbol{W}_{1,h}, \boldsymbol{W}_{2,h}\}$ denotes the parameters. Then the average Helmholtz free energy can be formalized as*

$$F^* = -\frac{1}{H} \sum_{h=1}^{H} T \log \sum_{i=1}^{N} e^{-\frac{\|\boldsymbol{W}_{1,h}\boldsymbol{z} - \boldsymbol{W}_{2,h}\boldsymbol{h}_i\|^2}{2T}}.$$

*Assuming that $\|\boldsymbol{W}_{1,h}\boldsymbol{z}\| = \|\boldsymbol{W}_{2,h}\boldsymbol{h}_i\| = \rho$ for all $i \in [N], h \in [H]$, the forward inference of the multi-head attention defined in Eq (11) can be modeled as one gradient descent step for minimizing $F^*$ with the learning rate $\eta$ when setting $\boldsymbol{W}_{Q,h}^T \boldsymbol{W}_{K,h} = \boldsymbol{W}_{1,h}^T \boldsymbol{W}_{2,h}$ and $\boldsymbol{W}_{O,h} \boldsymbol{W}_{V,h} = \frac{\eta T}{H} \boldsymbol{W}_{1,h}^T \boldsymbol{W}_{2,h}$ for all $h \in [H]$.*

*Proof.* Using the assumption that $\|\boldsymbol{W}_{1,h}\boldsymbol{z}\| = \|\boldsymbol{W}_{2,h}\boldsymbol{h}_i\| = \rho$ for all $i \in [N], h \in [H]$, we have

$$F^* = -\frac{1}{H} \sum_{h=1}^{H} T \log \sum_{i=1}^{N} e^{-\frac{\|\boldsymbol{W}_{1,h}\boldsymbol{z} - \boldsymbol{W}_{2,h}\boldsymbol{h}_i\|^2}{2T}} = \tilde{F}^* + \rho^2, \tag{19}$$

where $\tilde{F}^* = -\frac{1}{H} \sum_{h=1}^{H} T \log \sum_{i=1}^{N} e^{\frac{\boldsymbol{z}^T \boldsymbol{W}_{1,h}^T \boldsymbol{W}_{2,h} \boldsymbol{h}_i}{T}}$. We can take the derivative of $F$ with respect to $z$ to obtain

$$\nabla_{\boldsymbol{z}} F^* = \nabla_{\boldsymbol{z}} \tilde{F}^* = -\frac{T}{H} \sum_{h=1}^{H} \sum_{i=1}^{N} \frac{e^{\boldsymbol{z}^T \boldsymbol{W}_{1,h}^T \boldsymbol{W}_{2,h} \boldsymbol{h}_i / T}}{Z_h} \boldsymbol{W} \boldsymbol{h}_i, \tag{20}$$

where $Z_h = \sum_{j=1}^{N} e^{\boldsymbol{z}^T \boldsymbol{W}_{1,h}^T \boldsymbol{W}_{2,h} \boldsymbol{h}_j / T}$. Then, given an initial value $\boldsymbol{z}^{(0)}$, we can apply gradient descent to minimize the objective $\tilde{F}^*$. Suppose the learning rate is $\eta$, the iteration is given by

$$\boldsymbol{z}^{(k+1)} = \boldsymbol{z}^{(k)} - \eta_k \nabla_{\boldsymbol{z}^{(k)}} \tilde{F}^* = \boldsymbol{z}^{(k)} + \sum_{h=1}^{H} \sum_{i=1}^{N} \frac{e^{(\boldsymbol{z}^{(k)})^T \boldsymbol{W}_{1,h}^T \boldsymbol{W}_{2,h} \boldsymbol{h}_i / T}}{Z_h} \frac{\eta T}{H} \boldsymbol{W}_{1,h}^T \boldsymbol{W}_{2,h} \boldsymbol{h}_i. \tag{21}$$

Comparing with Eq (11), we can set $\boldsymbol{W}_{1,h}^T \boldsymbol{W}_{2,h} = \boldsymbol{W}_{Q,h}^T \boldsymbol{W}_{K,h}$ and $\boldsymbol{W}_{O,h} \boldsymbol{W}_{V,h} = \frac{\eta T}{H} \boldsymbol{W}_{Q,h}^T \boldsymbol{W}_{K,h}$ for $h = 1, \ldots, H$. Then, we will have $Z_h' = Z_h$ and the above equation can be reformulated as

$$\boldsymbol{z}^{(k+1)} = \text{MHA}(\boldsymbol{z}^{(k)}) = \boldsymbol{z}^{(k)} + \sum_{h=1}^{H} \sum_{i=1}^{N} \frac{e^{\boldsymbol{z}^T \boldsymbol{W}_{Q,h}^T \boldsymbol{W}_{K,h} \boldsymbol{h}_i / T}}{Z_h} \boldsymbol{W}_{O,h} \boldsymbol{W}_{V,h} \boldsymbol{h}_i, \tag{22}$$

which has the same form as Eq (11). Thus, we complete our proof. $\square$

### A.7 PROOF OF LEMMA 5

**Lemma 5.** *Both the Helmholtz free energy $F^*$ with respect to $\boldsymbol{z}$ and its upper bound $\tilde{F}^*$ are non-convex. Assume $\|\boldsymbol{W}_{1,h}\boldsymbol{z}\| \le \rho$ and $\|\boldsymbol{W}_{2,h}\boldsymbol{h}_i\| \le \rho$ for all $i \in [N]$ and $h \in [H]$. The local minima of $F^*$ are attained at the boundary $\|\boldsymbol{z}\| = \rho$ or when $\sum_{h=1}^{H} \sum_{i=1}^{N} p_{i,h} \boldsymbol{W}_{1,h}^T (\boldsymbol{W}_{1,h}\boldsymbol{z} - \boldsymbol{W}_{2,h}\boldsymbol{h}_i) = 0$ where $p_{i,h} = \frac{1}{Z_h} e^{-\frac{\|\boldsymbol{W}_{1,h}\boldsymbol{z} - \boldsymbol{W}_{2,h}\boldsymbol{h}_i\|^2}{2T}}$ and $Z_h = \sum_{i=1}^{N} e^{-\frac{\|\boldsymbol{W}_{1,h}\boldsymbol{z} - \boldsymbol{W}_{2,h}\boldsymbol{h}_i\|^2}{2T}}$. In addition, the local minima of $\tilde{F}^*$ are attained at the boundary $\|\boldsymbol{z}\| = \rho$.*

*Proof.* Recalling that $F^* = -\frac{1}{H} \sum_{h=1}^{H} T \log \sum_{i=1}^{N} e^{-\frac{\|\boldsymbol{W}_{1,h}\boldsymbol{z} - \boldsymbol{W}_{2,h}\boldsymbol{h}_i\|^2}{2T}}$. We compute the derivative of $F^*$ with respect to $\boldsymbol{z}$ as

$$\nabla_{\boldsymbol{z}} F^* = \frac{1}{H} \sum_{h=1}^{H} \sum_{i=1}^{N} p_{i,h} \boldsymbol{W}_{1,h}^T (\boldsymbol{W}_{1,h}\boldsymbol{z} - \boldsymbol{W}_{2,h}\boldsymbol{h}_i),$$

where $p_{i,h} = \frac{1}{Z_h} e^{-\frac{\|\boldsymbol{W}_{1,h}\boldsymbol{z} - \boldsymbol{W}_{2,h}\boldsymbol{h}_i\|^2}{2T}}$ and $Z_h = \sum_{i=1}^{N} e^{-\frac{\|\boldsymbol{W}_{1,h}\boldsymbol{z} - \boldsymbol{W}_{2,h}\boldsymbol{h}_i\|^2}{2T}}$. Since the attention heads are independent of each other, the proof for each head is similar to that of Lemma 2. We denote $\boldsymbol{r}_{i,h} = \boldsymbol{W}_{1,h}^T (\boldsymbol{W}_{1,h}\boldsymbol{z} - \boldsymbol{W}_{2,h}\boldsymbol{h}_i)$ and to compute the Hessian matrix, we first calculate

$$\nabla_{\boldsymbol{z}} p_{i,h} = -\frac{1}{T} p_{i,h} \boldsymbol{r}_{i,h} + \frac{1}{T} p_{i,h} \sum_{j=1}^{N} p_{j,h} \boldsymbol{r}_{j,h}.$$

Then the Hessian matrix of $F^*$ with respect to $\boldsymbol{z}$ is

$$
\nabla_{\boldsymbol{z}}^2 F^* = \frac{1}{H} \sum_{h=1}^{H} \left[ \sum_{i=1}^{N} \boldsymbol{r}_{i,h} \left( -\frac{1}{T} p_{i,h} \boldsymbol{r}_{i,h}^T + \frac{1}{T} p_{i,h} \sum_{j=1}^{N} p_{j,h} \boldsymbol{r}_{j,h}^T \right) + \boldsymbol{W}_{1,h}^T \boldsymbol{W}_{1,h} \right]
$$

$$
= \frac{1}{H} \sum_{h=1}^{H} \left[ \underbrace{\boldsymbol{W}_{1,h}^T \boldsymbol{W}_{1,h}}_{\succeq 0} - \underbrace{\frac{1}{T} \left( \sum_{i=1}^{N} p_{i,h} \boldsymbol{r}_{i,h} \boldsymbol{r}_{i,h}^T - \left( \nabla_{\boldsymbol{z}} F_h^* \right) \left( \nabla_{\boldsymbol{z}} F_h^* \right)^T \right)}_{\preceq 0} \right],
$$

where $F_h^*$ is the Helmholtz free energy in the $h$-th subspace and $\nabla_{\boldsymbol{z}} F_h^* = \sum_{i=1}^{N} p_{i,h} \boldsymbol{r}_{i,h}$. Therefore, we obtain that $F^*$ is neither convex nor concave and when $\|\boldsymbol{z}\| \leq \rho$, its local minima can only be attained at the boundary $\|\boldsymbol{z}\| = \rho$ or at interior points where $\nabla_{\boldsymbol{z}} F^* = 0$, that is, $\sum_{h=1}^{H} \sum_{i=1}^{N} p_{i,h} \left( \boldsymbol{W}_{1,h} \boldsymbol{z} - \boldsymbol{W}_{2,h} \boldsymbol{h}_i \right) = 0$. Similarly, we can obtain the Hessian matrix of $\tilde{F}^*$ as

$$
\nabla_{\boldsymbol{z}}^2 \tilde{F}^* = -\frac{1}{HT} \sum_{h=1}^{H} \left[ \sum_{i=1}^{N} p_{i,h} \boldsymbol{r}_{i,h} \boldsymbol{r}_{i,h}^T - \left( \nabla_{\boldsymbol{z}} \tilde{F}_h^* \right) \left( \nabla_{\boldsymbol{z}} \tilde{F}_h^* \right)^T \right] \preceq 0,
$$

where $p_{i,h} = \frac{e^{\boldsymbol{z}^T \boldsymbol{W}_{1,h}^T \boldsymbol{W}_{2,h} \boldsymbol{h}_i / T}}{Z_h}$ and $Z_h = \sum_{i=1}^{N} e^{\frac{\boldsymbol{z}^T \boldsymbol{W}_{1,h}^T \boldsymbol{W}_{2,h} \boldsymbol{h}_i}{T}}$. Therefore, we can get that $\tilde{F}^*$ is concave and when $\|\boldsymbol{z}\| \leq \rho$, its local minima can only be attained at the boundary $\|\boldsymbol{z}\| = \rho$. $\qquad\square$

## A.8 DETAILED DESIGN OF MHAtten2nd AND MHAtten2nd-1st

As in the single-head case, we extend the Newton's method-inspired modification of the attention structure to the multi-head setting. The update rule derived from the first-order gradient descent method for $F^*$ is

$$
\boldsymbol{z}^{(k+1)} = \boldsymbol{z}^{(k)} - \eta \nabla_{\boldsymbol{z}^{(k)}} F^* = \boldsymbol{z}^{(k)} - \frac{\eta}{H} \sum_{h=1}^{H} \sum_{i=1}^{N} p_{i,h} \boldsymbol{W}_{1,h}^T \left( \boldsymbol{W}_{1,h} \boldsymbol{z} - \boldsymbol{W}_{2,h} \boldsymbol{h}_i \right), \qquad (23)
$$

where $p_{i,h} = \frac{1}{Z_h} e^{-\frac{\|\boldsymbol{W}_{1,h} \boldsymbol{z} - \boldsymbol{W}_{2,h} \boldsymbol{h}_i\|^2}{2T}}$. The basic form using Newton's method based on second-order gradients is

$$
\boldsymbol{z}^{(k+1)} = \boldsymbol{z}^{(k)} - \eta \left[ \nabla_{\boldsymbol{z}^{(k)}}^2 F^* \right]^{-1} \nabla_{\boldsymbol{z}^{(k)}} F^*, \qquad (24)
$$

where $\left[ \nabla_{\boldsymbol{z}^{(k)}}^2 F^* \right]^{-1}$ is the Hessian matrix at $\boldsymbol{z}^{(k)}$. We denote the Helmholtz free energy in the $h$-th subspace as $F_h^* = -T \log \sum_{i=1}^{N} Z_h$ and then $F^* = \frac{1}{H} F_h^*$. Instead of applying Newton's method directly to $F^*$, we apply it independently to each subspace $F_h^*$, which can be formalized as

$$
\boldsymbol{z}^{(k+1)} = \boldsymbol{z}^{(k)} - \frac{\eta}{H} \sum_{h=1}^{H} \left[ \nabla_{\boldsymbol{z}^{(k)}}^2 F_h^* \right]^{-1} \nabla_{\boldsymbol{z}^{(k)}} F_h^* \qquad (25)
$$

Considering the analogous roles of $\boldsymbol{W}_{1,h}^T \boldsymbol{W}_{2,h}$ and $\boldsymbol{W}_{Q,h}^T \boldsymbol{W}_{K,h}$ in Theorem 2, we use the notation $\boldsymbol{q}_h = \boldsymbol{W}_{1,h} \boldsymbol{z}$, $\boldsymbol{k}_{i,h} = \boldsymbol{W}_{2,h} \boldsymbol{h}_i$ and $\bar{\boldsymbol{k}}_h = \sum_{i=1}^{N} p_{i,h} \boldsymbol{W}_{2,h} \boldsymbol{h}_i$. Then the Hessian matrix of $F_h^*$ can be formulated as

$$
\nabla_{\boldsymbol{z}}^2 F_h^* = \boldsymbol{W}_{1,h}^T \left[ \boldsymbol{I} - \frac{1}{T} \sum_{i=1}^{N} p_{i,h} \left( \boldsymbol{k}_{i,h} - \bar{\boldsymbol{k}}_h \right) \left( \boldsymbol{k}_{i,h} - \bar{\boldsymbol{k}}_h \right)^T \right] \boldsymbol{W}_{1,h}. \qquad (26)
$$

Note that due to $\boldsymbol{W}_{1,h} \in \mathbb{R}^{\frac{d}{H} \times d}$, the Hessian matrix $\nabla_{\boldsymbol{z}}^2 F_h^* \in \mathbb{R}^{d \times d}$ is non-invertible. Therefore, we employ the range-space approach in Newton's method, or equivalently, use the pseudoinverse[9] of the Hessian, i.e.,

$$
\left[ \nabla_{\boldsymbol{z}}^2 F_h^* \right]^{-1} = \boldsymbol{W}_{1,h}^T \left( \boldsymbol{W}_{1,h} \boldsymbol{W}_{1,h}^T \right)^{-1} \left[ \boldsymbol{I} - \frac{1}{T} \sum_{i=1}^{N} p_{i,h} \boldsymbol{d}_{i,h} \boldsymbol{d}_{i,h}^T \right]^{-1} \boldsymbol{W}_{1,h}, \qquad (27)
$$

---

[9] Here we use $\left( \boldsymbol{W}^T \boldsymbol{C} \boldsymbol{W} \right)^\dagger = \boldsymbol{W}^T \left( \boldsymbol{W} \boldsymbol{W}^T \right)^{-1} \boldsymbol{C}^{-1} \boldsymbol{W}$ when $\boldsymbol{W} \in \mathbb{R}^{m \times n}$ and $m < n$.

where we use $\boldsymbol{d}_{i,h} = \boldsymbol{k}_{i,h} - \bar{\boldsymbol{k}}_h$ for simplicity. Furthermore, by parameterize $\boldsymbol{W}_{1,h}, \boldsymbol{W}_{2,h}$ as $\boldsymbol{W}_{Q,h}, \boldsymbol{W}_{K,h}$, the $\mathrm{Atten2nd}(\boldsymbol{z})$ can be extended as

$$\mathrm{MHA2nd}(\boldsymbol{z}) = \boldsymbol{z} + \frac{\eta}{H} \sum_{h=1}^{H} \boldsymbol{P}_h \left( \boldsymbol{q}_h - \bar{\boldsymbol{k}}_h \right),$$

$$\boldsymbol{P}_h = \boldsymbol{W}_{Q,h}^T \left( \boldsymbol{W}_{Q,h} \boldsymbol{W}_{Q,h}^T \right)^{-1} \left[ \boldsymbol{I} - \frac{1}{T} \sum_{i=1}^{N} p_{i,h} \boldsymbol{d}_{i,h} \boldsymbol{d}_{i,h}^T \right]^{-1} \boldsymbol{W}_{Q,h} \boldsymbol{W}_{Q,h}^T. \tag{28}$$

Below, we first consider the computational cost for a single head. The cost to compute $\boldsymbol{q}_h - \bar{\boldsymbol{k}}_h$ and all $\boldsymbol{d}_{i,h}$ is $O(\frac{Nd}{H} + \frac{d^2}{H})$. It should be noted that $\boldsymbol{W}_{Q,h} \boldsymbol{W}_{Q,h}^T$ and its inverse only need to be pre-computed once and therefore the cost can be ignored when generating a large number of tokens. The cost of computing the outer products of $N$ vectors and the inverse are $O(N\frac{d^2}{H^2} + \frac{d^3}{H^3})$. And performing the remaining matrix multiplications need $O(\frac{d^2}{H^2} + \frac{d^2}{H})$. Thus the total cost for one head is $O(N\frac{d^2}{H^2} + \frac{d^2}{H} + \frac{d^3}{H^3})$. Considering there are $H$ heads, the final cost is $O(Nd\frac{d}{H} + d^2 + d^2\frac{d}{H^2})$. Compared with $O(Nd + d^2)$ of standard attention, this incurs a higher computational cost.

To reduce the computational cost, as in the previous case, we replace the matrix inversion with the first-order Taylor expansion, which can be formalized as

$$\mathrm{MHA2nd1st}(\boldsymbol{z}) = \boldsymbol{z} + \frac{\eta}{H} \sum_{h=1}^{H} \boldsymbol{P}_h \left( \boldsymbol{q}_h - \bar{\boldsymbol{k}}_h \right),$$

$$\boldsymbol{P}_h = \boldsymbol{W}_{Q,h}^T \left( \boldsymbol{W}_{Q,h} \boldsymbol{W}_{Q,h}^T \right)^{-1} \left[ \boldsymbol{I} + \frac{1}{T} \sum_{i=1}^{N} p_{i,h} \boldsymbol{d}_{i,h} \boldsymbol{d}_{i,h}^T \right] \boldsymbol{W}_{Q,h} \boldsymbol{W}_{Q,h}^T. \tag{29}$$

In fact, this can be further simplified as

$$\mathrm{MHA2nd1st}(\boldsymbol{z}) = \boldsymbol{z} + \frac{\eta}{H} \sum_{h=1}^{H} \boldsymbol{W}_{Q,h}^T \left[ \left( \boldsymbol{q}_h - \bar{\boldsymbol{k}}_h \right) + \boldsymbol{b}_h \right],$$

$$\boldsymbol{b}_h = \left( \boldsymbol{W}_{Q,h} \boldsymbol{W}_{Q,h}^T \right)^{-1} \frac{1}{T} \sum_{i=1}^{N} p_{i,h} \boldsymbol{d}_{i,h} \left[ \boldsymbol{d}_{i,h}^T \boldsymbol{W}_{Q,h} \boldsymbol{W}_{Q,h}^T \left( \boldsymbol{q}_h - \bar{\boldsymbol{k}}_h \right) \right]. \tag{30}$$

In this case, the cost to compute $\boldsymbol{q}_h - \bar{\boldsymbol{k}}_h$ and all $d_{i,h}$ is still $O(\frac{Nd}{H} + \frac{d^2}{H})$. However, computing $\boldsymbol{b}_h$ only needs $O(\frac{d^2}{H} + \frac{Nd}{H} + \frac{d^2}{H^2})$ by prioritizing the computation of inner products between vectors. Finally, the remaining cost of matrix multiplication is $O(\frac{d^2}{H})$. Therefore, the cost for each head is $O(\frac{Nd}{H} + \frac{d^2}{H})$ and the total cost for $H$ heads is $O(Nd + d^2)$, which is of the same order as standard attention.

In practice, to avoid additionally computing and storing $d_{i,h}$, we adopt the following form.

$$\mathrm{MHA2nd1st}(\boldsymbol{z}) = \boldsymbol{z} + \frac{\eta}{H} \sum_{h=1}^{H} \boldsymbol{W}_{Q,h}^T \left[ \left( \boldsymbol{q}_h - \bar{\boldsymbol{k}}_h \right) + \boldsymbol{b}_h \right],$$

$$\boldsymbol{b}_h = \left( \boldsymbol{W}_{Q,h} \boldsymbol{W}_{Q,h}^T \right)^{-1} \frac{1}{T} \left[ \sum_{i=1}^{N} p_{i,h} \boldsymbol{k}_{i,h} \left( \boldsymbol{k}_{i,h}^T \boldsymbol{u}_h \right) - \bar{\boldsymbol{k}}_h \left( \bar{\boldsymbol{k}}_h^T \boldsymbol{u}_h \right) \right].$$

$$\boldsymbol{u}_h = \boldsymbol{W}_{Q,h} \boldsymbol{W}_{Q,h}^T \left( \boldsymbol{q}_h - \bar{\boldsymbol{k}}_h \right) \tag{31}$$

In practice, we also introduce new parameters $\boldsymbol{W}_O \in \mathbb{R}^{d \times d_h}$ to replace $\frac{\eta}{H} \boldsymbol{W}_{Q,h}^T$ to make the model more flexible. Moreover, to maintain stability, we set the temperature $T$ in the attention score $p_{i,h}$ as a head-wise learnable parameter with initialization as $\boldsymbol{d}_h$ and the temperature in $\boldsymbol{b}_h$ is treated in the same way.

## A.9 MORE DETAILS OF EXPERIMENTS

We mainly follow the setup of Feng et al. (2024); Yang et al. (2024a). For the LIS task, we investigate different task lengths $L = \{10, 20, 30, 40\}$ which denotes the length of the input sequence to solve. For each problem size, the training and test sets were generated independently with sizes of 51,200 and 5,120 respectively. We uniformly set the batch size to 128. The model dimensions is selected from $d = \{32, 64, 128, 256\}$ and the number of layers is set to 3 by default for all models. We use a fixed dropout ratio of 0.1 for all experiments to improve generalization. For positional encoding, we use the absolute positional encoding as in Vaswani et al. (2017). All models are trained for 300 epochs using AdamW(Loshchilov, 2017) with with $\beta_1 = 0.9$, $\beta_2 = 0.999$, lr $= 1e - 4$ and weight decay of 0.01. During training, the model is optimized using cross-entropy loss on the answer tokens, while a greedy decoding strategy is employed during testing. For the results presented in the form of heat maps, we report the average test accuracy over the last five epochs as the final accuracy. Furthermore, our experiments were conducted on four 24GB NVIDIA GeForce RTX 3090 GPUs and can be completed within two days.

For more experimental results, we present Figure 2 the test accuracy under different task difficulties and model sizes under the attention-only configuration and the configuration incorporating MLP. In Figures 3 and 4, we show the test accuracy of MHA(-only) and MHA2nd1st(-only) during training.



Figure 2: Test accuracy on LIS tasks across different task lengths and model sizes. **Left part:** The accuracy of MHA-only and MHA2nd1st-only. **Right part:** The accuracy of MHA and MHA2nd1st.

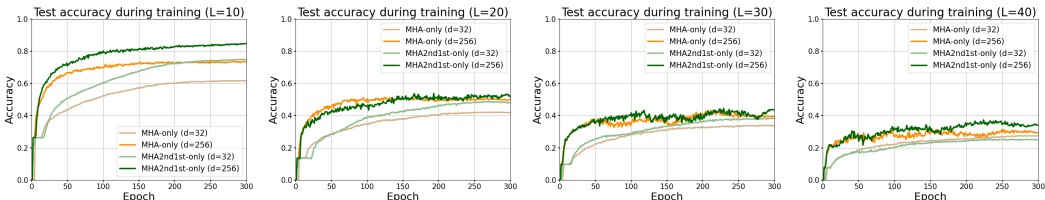

Figure 3: Test accuracy on LIS tasks of MHA-only and MHA2nd1st-only during training when the task length $L = \{10, 20, 30, 40\}$ and the model dimension $d = 32/256$.

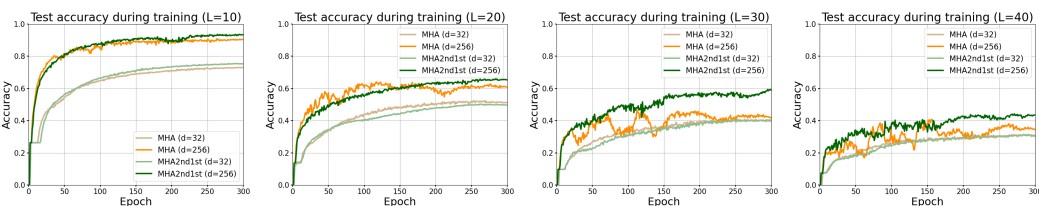

Figure 4: Test accuracy on LIS tasks of MHA and MHA2nd1st during training when the task length $L = \{10, 20, 30, 40\}$ and the model dimension $d = 32/256$.

