# OpenReview forum: "Transformers as Intrinsic Optimizers: Forward Inference through the Energy Principle"
_ICLR.cc/2026/Conference — ICLR 2026 Conference Withdrawn Submission_

### Official Review · Reviewer_Pydm · 2025-10-29

**Soundness:** 3
**Presentation:** 4
**Contribution:** 3
**Rating:** 6
**Confidence:** 2

**Summary:**

This paper revisits the energy perspective to understand attention-based transformers, and proposes an interpretation where attention computation and weight updates are jointly applying alternative minimization of the Helmholtz free energy. Moreover, based on this interpretation, the authors propose a novel attention architecture based on inspired by Newton's method to minimize the free energy. Finally, experimental results on the *Longest Increasing Subsequence task* show potential advantage of using this novel architecture compared with standard attention architecture.

**Strengths:**

**Novel unified optimization perspective:** this paper proposes a novel unified framework to interpret the forward inference and backward inference as jointly minimizing the free energy using alternating gradient descent, bridging the gap between energy-based modeling and deep network dynamics.

**Theory inspired algorithm** The motivation behind the proposed architectures is clear since it arises naturally from the theoretical derivation in Section 4. Moreover, the idea of preconditioning token updates using local covariance is intuitive and beautifully arise from the mathematical derivations.

**Clear writing and sufficient background** This paper is very-well written with presentation of the background on energy-based framework for transformers, and connections to both EBMs. Moreover, all the theorems and notations are clearly defined.

**Weaknesses:**

**Preliminary empirical validation** The experiments only focus on *Longest Increasing Subsequence task* which is a bit weak. It would be better to see how the proposed transformers perform on more realistic language tasks.

**Strong assumptions on Theorem1** The interpretation and Theorem 1 rely on the assumption that $\|\|z\|\|=\|\|Wh\|\|$ and $\eta T W_Q^\top W_K=W_V$. This assumption seems to only hold with layer normalization. Can the authors provide some other examples where this assumption holds?

**Insufficient comparison with prior work** This paper lacks the comparisons of their results, such as interpretation of transformers as free energy minimization, alternating minimization between forward and backward inference, and proposed new architecture, with prior work, such as [1]

[1] White-box transformers via sparse rate reduction.

[2] Attention-only transformers via unrolled subspace denoising.

**Questions:**

Please see weakness.

---

### Official Review · Reviewer_L3AG · 2025-11-01

**Soundness:** 2
**Presentation:** 3
**Contribution:** 2
**Rating:** 2
**Confidence:** 2

**Summary:**

The paper proposes an energy-based interpretation of Transformer attention, viewing the forward pass as a single gradient descent step on a Helmholtz free energy function defined with an elastic potential energy. Under this view, query–key interactions correspond to minimizing energy, and residual connections represent incremental optimization updates. The authors further extend the formulation to a second-order (Newton-style) attention update and introduce a practical first-order approximation (Att2nd1st) that retains the same asymptotic complexity as standard attention. They also generalize the framework to multi-head attention (MHA2nd1st) and conduct experiments on the Longest Increasing Subsequence (LIS) task, showing modest performance gains over standard attention.

**Strengths:**

The paper presents a clear and unified theoretical framework connecting Transformer attention mechanisms with energy minimization principles. The Helmholtz free energy perspective provides an interpretable physical analogy for attention updates. The derivation of both first- and second-order variants is mathematically sound and highlights the potential for curvature-aware improvements. The multi-head extension is conceptually consistent and technically elegant. The presentation is clear, with good organization and readable figures.

**Weaknesses:**

1. **Limited experimental scope**
 The paper validates the proposed framework only on a synthetic task (Longest Increasing Subsequence, LIS). It lacks evaluations on realistic NLP or vision benchmarks such as language modeling or image classification, leaving the practical effectiveness and generalization ability unclear.

2. **Small performance gains**
 The improvements reported on the LIS task are modest, without statistical significance analysis or comparisons against stronger baselines (e.g., efficient attention variants, long-range reasoning tasks). This weakens the empirical support for the claimed advantages.

3. **Unquantified computational overhead**
 Although the authors claim asymptotic equivalence in complexity to standard attention, the paper does not measure inference latency, memory usage, FLOPs, or throughput. Constant-factor overheads from additional inner products and covariance computations could hinder real-world efficiency.

4. **Idealized theoretical assumptions**
 The derivation relies on norm constraints and simplified conditions that may not hold in practice, particularly in transformers with LayerNorm. The paper does not empirically test how these assumptions align with real network behavior.

**Questions:**

Please refer to the weakness section

---

### Official Review · Reviewer_Fjgf · 2025-11-02

**Soundness:** 1
**Presentation:** 2
**Contribution:** 2
**Rating:** 2
**Confidence:** 5

**Summary:**

The authors formulate **attention as an energy-minimization process**. There are, however, **key gaps**: To apply their attention scheme, they use a second-order update (Hessian-based). However, they also prove that the energy surface is non-convex. The Newton-style second-order *updates can therefore be unstable*, and the update may nudge the system in the direction of non-optimal or divergent regions. This is not addressed. The authors introduce a learnable temperature parameter as a practical stability knob, but they *do not present any formal stability analysis*. Also, to reduce computational cost, the authors do not use the full Hessian inverse. Instead, they approximate using a Taylor/Neumann series where they only keep the first term. The paper provides *no analysis of when this approximation is valid* and whether there are cases where it can become problematic.

The authors show that **dot-product attention is a special case of their proposed energy-based formulation**, but only **under very rigid constraints**. In particular, they assume *tight parameter tying between the query, key, and value projection matrices*. The derivation is quite elegant, but it is largely *disconnected from how real transformers work*. The authors do acknowledge that these rigid constraints can be relaxed through normalization. However, this exercise still presents only a *conceptual analogy without providing a practically useful insight*.

In the **experimental validation section there are key gaps**: the *paper does not mention including the Helmholtz free-energy regularization term in the training objective*. There is *no mention of alternating updates over the latent representations and model parameters* (the core of their theoretical “alternating optimization” method). Instead, the *text describes only a direct training procedure:* a transformer model with energy-based attention layers trained in the same way as a conventional transformer. The authors *do not explicitly explain why the free-energy term was omitted* from the experiments. They *do not discuss whether including it would have been computationally difficult*, unstable, or simply beyond the paper’s experimental scope. They also *don’t include any ablation* to show how this omission might affect the relationship between their theory and results. This is a significant omission from the paper that is simply not rationalized.

In general, the **empirical evaluation in the paper is too narrow to convincingly demonstrate the usefulness of the proposed attention mechanism**. All of the *experiments are conducted on the Longest Increasing Subsequence (LIS) task only*, which is a synthetic algorithmic reasoning benchmark. While such a task can be useful for an initial proof-of-concept validation, the proposed method should be more rigorously evaluated on language, vision, or even multimodal datasets. Therefore, the *current results do not provide evidence that the proposed energy-based attention generalizes beyond a small toy problem*. As presented, the evaluation *does not conclusively show whether the method scales to realistic sequence lengths, maintains computational efficiency under typical transformer workloads, or offers measurable benefits on any of the established benchmarks (such as GLUE, Long Range Arena, ImageNet, etc.).*

Although, I highly appreciate the innovativeness of the proposed approach, without results on established benchmarks, currently, the experimental section reads more like an early conceptual demonstration than a rigorous evaluation of a new transformer architecture.
The paper cites prior work that interprets transformers through an energy-based or Hopfield-like framework. However, it **does not empirically compare its proposed method** to either of these models. The authors acknowledge these studies in the appendix and note that both established a formal equivalence between attention mechanisms and energy minimization processes. However, the experimental section compared the proposed architecture to the standard baseline architecture only. This omission leaves a gap in the empirical validation since the paper’s main claim is that it provides a more general energy-based framework that extends prior work. To prove this claim, the authors would need to show why this new formulation performs better (in accuracy, convergence speed, stability, or computational efficiency) compared to the existing energy-based transformer architectures. However, the paper in its current form, fails to achieve that.

**Strengths:**

The paper formulates **attention as an energy-minimization process**. This approach is **innovative,  and some of the derivations are elegant**.

The paper also **shows that dot-product attention is a special case of their proposed energy-based formulation**. It's always nice to see new perspectives on such an important mechanism of today.

The paper also **attempts to empirically validate their energy-minimization approach**, with at least preliminary support.

**Weaknesses:**

Unfortunately, as detailed in the summary:

**The theoretical/design formulation has key gaps** : *stability issues* with the proposed optimization, *no stability analysis for the temperature parameter,* and *lacking analysis for soundness of approximations* used.

**Limited applicability of the link with dot-product attention**:  Dot-product attention is shown as a special case of the proposed energy-based formulation *only under very rigid constraints*. Although it's mentioned these could be relaxed via normalization, the link presented thus feels somewhat preliminary for ICLR--nice intuition but potentially of limited impact--especially since design applications and strong empirical justification are both lacking.

**Unconvincing experimental work**: the experimental validation section has important gaps as outlined in the corresponding paragraph of the summary above. Namely, the *focus on the LIS task* only is quite restrictive; the evaluation *does not conclusively show whether the method scales to realistic sequence lengths, maintains computational efficiency under typical transformer workloads, or offers measurable benefits on any of the established benchmarks*; the authors fail to show *why their new formulation performs better (in accuracy, convergence speed, stability, or computational efficiency) compared to existing energy-based transformer architectures*.

**Questions:**

Do the authors dispute any of the technical points that I've listed above as weaknesses? (see the 7 points in italics under Weaknesses and their more detailed discussion in Summary)

Or do the authors agree with the technical points but just disagree that those issues make the paper in its present form too weak for ICLR?

---

### Official Review · Reviewer_f8WA · 2025-11-04

**Soundness:** 2
**Presentation:** 3
**Contribution:** 2
**Rating:** 4
**Confidence:** 3

**Summary:**

This paper proposes a framework that explains the forward process of transformers (specifically, a single layer self-attention)  as the optimization process of a Helmholtz free energy.

**Strengths:**

- The proposed framework is well presented with a clear derivation.
- The proposed energy minimization perspective is integrated with the backward process through an alternating optimization process, which can be helpful to use the energy minimization perspective to understand the training process of transformers.

**Weaknesses:**

- There have been many works that use the energy minimization perspective to understand the forward process of transformers (e.g. [1]). The authors mentioned that " Although these studies establish certain connections between energy and Transformers, the design of energy functions is often not straightforward and lacks a unified framework to understand...". However, I don't see how this work is more straightforward or unified than the existing ones.
- In the derivation of the energy function, the authors assumed $h_i$ and $z$ are separate variables (i.e. when you calculate the gradient of $z$, the $h_i$ is viewed as a constant). However, in self-attention, a token can also attend to itself, which basically means there can be an $i_0$ such that $h_{i_0} = z$. I wonder how the proposed framework handles this situation.
- The title is kind of over-claiming. This paper only considers a single-layer self-attention, while there are still a lot of challenges to be solved in extending this perspective to a full transformer model. Actually, a previous work [2] proposed four major challenges of explaining transformers through an energy minimization perspective, and in my opinion, this paper has only solved the first one. (I'm not sure if this paper solved the fourth challenge, i.e. the asymmetry issue. I don't see any symmetry requirements on $W$, but the authors also don't mention this)
- Regarding the alternating optimization part, how can you recover the forward process from (6), since the variable $z$ exists in both the loss term (the CE term) and the regularization term (the $F^\*$ term). In my understanding, the forward process only comes from the $F^\*$ term, but when you optimize $z$, you need to calculate the gradient from both terms, won't that break the explanation of the forward process?
- The experiments are kind of weak, and I don't see any verifications of the theoretical results. For example, can you do an experiment that verifies if the energy does go down when you do self-attention?

Minor issues:
Some mathematical equations are vague. For example, in Lemma 1, when you say "The system's free energy defined by Eq (2) attains its minimum value", what is this minimum value with respect to (i.e. what is the optimization variable when you say "minimum value")?

[1] Hopfield Networks is All You Need

[2] Transformers from an Optimization Perspective

**Questions:**

See Weaknesses.

---

### Note · Authors · 2025-12-03

**Comment:**

We sincerely thank the reviewers for their valuable time and effort in reviewing our manuscript. Their constructive feedback has been incredibly helpful, and we will carefully consider all the suggestions provided in order to strengthen and improve our work. After thoughtful reflection, we have decided to withdraw our submission at this stage. We plan to incorporate their feedback and make the necessary revisions to enhance the presentation of our research. We look forward to sharing the improved version with the community in the near future and truly appreciate the support and understanding of the reviewers and ACs.

**Withdrawal Confirmation:**

I have read and agree with the venue's withdrawal policy on behalf of myself and my co-authors.